# Habitat-specific trends in taxonomic, functional, and phylogenetic diversity in European plant communities over a century

Despite widespread concern over global biodiversity loss, the balance between gains and losses within local plant communities remains contentious, largely due to a scarcity of integrative, long-term and large-scale analyses across different habitats and multiple facets of biodiversity. Here, we analyse 57,390 vegetation-plot time series of vascular plants across Europe to quantify the average and habitat-specific trends in taxonomic, functional, phylogenetic, and gamma diversity, alongside with changes in threatened Red List, non-native, and specialist versus generalist species. We find that, over the last 100 years, plant communities gained on average 0.7% in vegetation cover and 0.2% in species number per year, associated with gains in functional and phylogenetic diversity, non-native, Red List, and generalist species. Diversity changes are most pronounced in mire and wetland communities. Differences among habitat types and habitat-change trajectory (stable, successional, disturbed), together with the most recent observation year, explain 2.1%–36.6% of the variation in diversity trends. Habitat-specific gamma diversity showed no general trends and only increased in stable grasslands and successional sparsely vegetated habitats. By integrating habitat types and change trajectories, we reconcile some of the conflicting narratives on local biodiversity change in favour of a more nuanced understanding of the observed variation in local biodiversity change.

Losses of plant diversity can alter the health and functioning of ecosystems, with negative effects also for human well-being[1,2]. Yet, plant diversity is threatened worldwide due to land-use change, climate change, environmental pollution, and biological invasions[2–7]. Beyond these pervasive threats to global plant diversity, the resulting changes in the diversity of local plant communities remain ambiguous and poorly understood[8,9]. One likely reason for the inconsistent findings across studies on local biodiversity changes is that the diversity trends of local plant communities are often context-dependent and influenced by the degree of specialisation among co-occurring species[8,10–16]. Another reason for the varying and non-significant local diversity trends could be that declines in habitat-specialist "loser" species may locally be masked by increases in more generalist "winner" species[15–18].

Regardless of the exact causes, the trends in local plant diversity changes remain debated. This is the case not only for taxonomic diversity (i.e., the number and evenness of species or taxa within a community), but also the diversity of plant forms and functions (i.e., the types and distribution of plant traits, Supplementary Table 1), the diversity of evolutionary histories (i.e., the phylogenetic relatedness of co-occurring taxa), as well as the number and cover of threatened Red List species, introduced non-native species, and habitat specialists versus generalist species (Fig. 1). Although global patterns in plant taxonomic, functional, and phylogenetic diversity are known to be complementary and linked to human well-being[2,19,20], we still lack integrative syntheses on the prevailing changes in local plant diversity at the continental and global extent.

✉ e-mail: stephan.kambach@gmail.com

**Fig. 1 | Average annual percentage changes, relative to baseline conditions, in local plant diversity indices across all time series.** Numbers in boxes show weighted average trends in percent per year. Dotted boxes indicate non-significant trends from separate Student's *t* tests at *p* > 0.05 (c.f., Supplementary Table 3). Horizontal lines below boxes show corresponding Wald-approximated 95% confidence intervals, calculated as weighted average ± 1.96 * standard error. Plant traits used to calculate functional diversity indices: plant height, stem diameter, rooting depth, specific root length, specific fine root length, leaf carbon to nitrogen ratio, leaf phosphorus content, leaf dry matter content, leaf area, leaf thickness, specific leaf area, stem conduit density, stem conduit diameter, and seed mass. Symbols: $i$ – focal species, $p_i$ – proportion of focal species, $n_i$ – number of focal species, $S$ – number of species, $X_{tS}$ – trait × species matrix, $PEW_{ij}$ – partial weighed evenness between species $i$ and $j$, $\Delta d$ – sum of abundance-weighted deviances, $\overline{dG}$ – mean distance of all species to the centre of trait gravity, $\Delta|d|$ – absolute abundance-weighted deviances from the centre of gravity, $b_i$ – phylogenetic branch length of focal species, $d_{ij}$ – phylogenetic distance between species $i$ and $j$.

One of the most robust methods for assessing temporal changes in local plant communities relies on repeated observations in (semi-) permanent vegetation plots[21,22]. These vegetation-plot time series are typically collected for national-level biodiversity monitoring programmes, e.g. across France, Switzerland, and the United Kingdom[23–25]. However, large-scale and generalizable syntheses across vegetation-plot time series are still often limited in terms of spatial and temporal representativeness (at least for the previous century), and in terms of differences among habitat types and specific drivers of biodiversity changes[26,27]. To foster European syntheses on local plant diversity changes across multiple regions, environmental gradients, and habitat types[21,28–30], we mobilized existing vegetation-plot time series data across large spatiotemporal extents[30–33]. The accumulated coverage in space and time now allows us to relate observed local and habitat-specific trends in plant diversity to the European-level richness of plant species, hereafter habitat-specific European gamma diversity[34–36].

To quantify the variability in diversity trends among different habitat types, all recorded vegetation-plot observations should ideally be classified into a common system of plant taxonomy and habitat types. Expert-based classification algorithms can nowadays combine information on the location and floristic composition to classify thousands of vegetation-plot observations into a standardized habitat system. For this study, we used the European Nature Information System (EUNIS, Supplementary Data 1, Supplementary Fig. 1), which is also adopted by the European Environmental Agency, ranging from broadly defined EUNIS level 1 habitats (e.g., T - forests) to more narrowly defined level 2 habitats (e.g., T1 - Deciduous broadleaved forest) and level 3 habitats (e.g., T18 - *Fagus* forest on acid soils)[14]. By assigning each vegetation-plot observation to a habitat type, we can not only account for the initial habitat type but also for the shifts from one habitat type to another within individual time series. For example, a secondary grassland that undergoes succession may eventually develop into a shrubland and later into a forest, both of which would be reflected in predictable shifts in species compositions. Conversely, forests that suffer from severe disturbance (e.g., storms, fires, or land-use changes) may develop into shrublands, grasslands, or more sparsely vegetated habitats. Even within relatively stable time series (e.g.,

plots that remain forested), a change of climate, pollution, or management practices may lead to likewise predictable shifts in plant composition[37,38].

In this study, we combined a large European dataset of vegetation-plot time series (using 57,390 time series with 199,282 individual vegetation-plot observations from ReSurveyEurope[33], Fig. 3a, b, Supplementary Fig. 2, and Supplementary Fig. 3), with data on plant traits[39], phylogenetic relationships[40] (Supplementary Data 2), threat status from static national and European Red Lists (providing a consistent classification across species and time)[41], non-native origin[42], and species' niche width derived from co-occurrence data[43–45] (Fig. 2). Most time series consist of two (55%), three (17.7%), or four (8.6%) observations and cover 1–4 (10.3%), 5–10 (43.1%), and 11–20 (18.1%) years between the first and the last observation date (with a maximum of 103 years, Supplementary Fig. 2). Based on the assigned level 3 habitat types at the first versus last observation date, we developed an expert-based scheme to classify the observed shifts in species composition into a more generalizable framework with the following five categories (all shown in the Supplementary Data 3). Time series with stable trajectories have no or only marginal shifts in EUNIS level 3 habitat types (13,527 = 23.6%). Time series with successional trajectories exhibit shifts from lower to higher vegetation biomass or complexity (3263 = 5.7%). Time series with disturbance trajectories show shifts that were contrary to successional dynamics (1945 = 3.4%). Time series with other trajectories indicate undirected shifts, changes in abiotic conditions, or land-use shifts (10,819 = 18.9%), and unclassifiable time series could not be assigned to EUNIS level 3 habitat types (27,836 = 48.5%). With this comprehensive dataset, we tested three hypotheses: (H1) local plant communities show stable species richness, but negative trends in functional and phylogenetic diversity due to the replacement of distinct habitat-specialist with more similar generalist species[5,46,47]. (H2) Diversity trends differ with respect to habitat types, habitat-change trajectories, and observation dates. (H3) Across all vegetation-plot observations, European gamma diversity has increased during the last 100 years due to a spread of non-native and generalist species. As disturbed and nutrient-rich habitats tend to be more susceptible to invasive species[48,49], we expected that time series

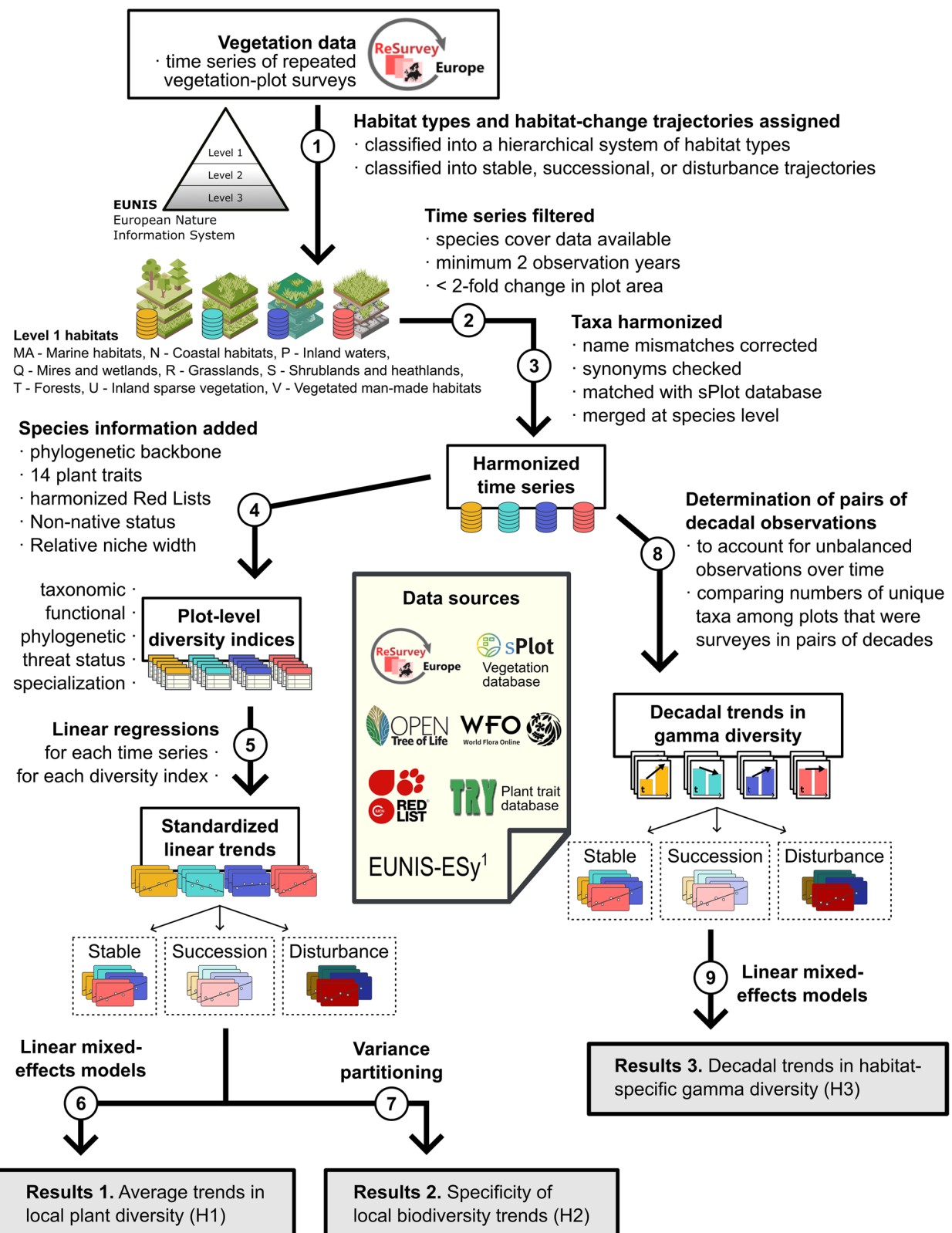

**Fig. 2 | Workflow for the compilation, cleaning, harmonization, and analysis of ReSurveyEurope time series of vegetation plots.** Different colours indicate separate analyses for different habitat types and habitat-change trajectories. Designed by FreePik. WFO-Logo: World Flora Online (2026). Published on the Internet, www.worldfloraonline.org.[1] EUNIS-ESy: Expert system for automatic classification of European vegetation plots to EUNIS habitats[91].

in disturbed and man-made habitats should show the comparatively largest increases in habitat-specific gamma diversity. With these analyses, we provide a comprehensive synthesis of the trends in local plant diversity across Europe, with important insights into the habitat-specificity of multiple facets of plant-diversity change. Adding to previous syntheses, we provide a nuanced understanding of the past and current changes in plant diversity that might serve as a blueprint for analyses in other biogeographic regions.

## Results and discussion

### Average trends in local plant diversity

To be able to draw generalizations from time series with varying numbers of vegetation plots and observations, we used a linear model-based, meta-analytic approach to summarise the biodiversity trend of each time series into a metric of mean annual diversity change. Averaged across all time series, we found that local plant communities exhibited positive annual percentage changes (relative to the first observation date) in nearly all of the investigated biodiversity indices (Fig. 1, Supplementary Table 2), such as vegetation cover (0.7% year⁻¹), species richness (0.2% year⁻¹), functional and phylogenetic diversity (0.1–0.7% year⁻¹), as well as in the number and cover of non-native, generalist, and threatened Red List species. Functional evenness and the number and cover of habitat specialists were the only indices that showed no significant trends. Considering these findings, we have to reject our hypotheses of a generally stable or decreasing local plant diversity in Europe (H1). For the diversity of threatened species, we want to point out that the estimated trends were not driven by changes in threat classification (as these were derived from static Red Lists), and we noted that threatened species tended to have slightly narrower niche widths than non-threatened species ($R^2 = 0.006$, Supplementary Box 1). When analysed on the original unit of the respective diversity indices, we estimated positive average annual percentage changes in mean nearest taxon distance, cover of non-native species, cover of habitat generalists, and community-weighted mean niche width, together with negative average trends in Shannon diversity, functional evenness, number and cover of Red List species, number of specialists, and cover of generalists (Supplementary Table 3).

To account for the uneven representation of habitat types, we recalculated all average diversity trends for the subset of EUNIS level 3-assigned time series, which we weighted to ensure an equal impact for all habitat types included. The obtained balanced annual percentage changes were of comparable significance and direction as those obtained with unbalanced analyses across all time series (Supplementary Table 4), yielding positive balanced trends for vegetation cover (1% year⁻¹), species richness (1.4% year⁻¹), and the other indices of plant diversity, except for non-significant balanced trends in functional divergence, number/cover of threatened Red List species and number of specialist species. When analysed on their original units, the significance of the resulting balanced trends was similar to those in annual percentage changes, albeit with a significant negative trend in the number/cover of threatened Red List species and non-significant trends in the number/cover of non-native species (Supplementary Table 5).

Addressing potential impacts of dataset biases, we found that the observed diversity trends were unrelated to rare changes in plot sizes within time series (Supplementary Table 6). Yet, compared to a permanent plot design, we found that semi-permanent vegetation plots were associated with more positive trends in species richness and number/cover of non-native species, together with more negative trends in Shannon diversity, mean pairwise phylogenetic distance, mean nearest taxon distance, number/cover of threatened Red List species, and more negative trends in the cover of specialist species (Supplementary Table 6). Motivated by recent findings of temporal shifts from negative to positive trend in European local species richness[50–52], we tested whether average annual percentage changes in local plant diversity differed between observations made before versus after 2000. Summarized among grassland, shrubland, and forests habitats, we found that temporal differences appeared between grassland and forest but were nonsignificant for shrubland (Supplementary Fig. 4, statistics in Supplementary Data 4). In contrast to Midolo et al.[52], plot observations conducted after 2000 yielded less positive trends, e.g., for vegetation cover of grasslands and shrublands, species richness of grasslands, and for the number/cover of threatened Red List species. Only the cover of habitat specialists in grasslands and forests showed more positive trends after 2000.

### Shifts in habitat types

For the subset of time series that could be assigned to EUNIS level 1 habitat types at the initial and the last observation date (see Methods), we noted that shifts from grasslands to shrublands and heathlands (and vice versa) were the most often recorded shift in EUNIS level 1 habitat types (in 715 and 449 time series, respectively, Fig. 3c, Supplementary Figs. 5 and 6). In contrast, any shifts from other habitat types towards sparsely vegetated or man-made habitats, both interpreted as disturbance trajectories, were among the least often recorded shifts in our dataset.

For a subset of 15,348 time series with sufficient replication of EUNIS level 3 habitat types and habitat-change trajectories (see Methods), we estimated the proportion of biodiversity trends variability that could be related to (1) different habitat types (EUNIS level 3 at the initial observation date), (2) different habitat-change trajectories (stable, succession, or disturbance), (3) interactions among habitat types and change trajectories (to account for habitat-change specificity), and (4) different observation periods (here, quantified by the year of the last observation to account for potential temporal gradients in climate, land-use, or scientific focus). Jointly, these four predictors explained between 2.1% and 36.6% of the variation in annual percentage changes of biodiversity indices (with a mean of 7.4%). In accordance with our hypothesis (H2), we found that the proportion of explained variation in diversity trends was, on average, most tightly related to interactions among level 3 habitat type and habitat-change trajectory (3.6% of explained variation, Fig. 4, statistics in Supplementary Data 5), followed by differences among level 3 habitat types (1%), observation years (0.7%), and the type of habitat-change trajectory (0.6%). The high importance of habitat type × change interactions, together with the modest explanatory power, highlights the specificity of local biodiversity patterns that must be addressed with further multi-scale contextual data (e.g., on land-use intensity, as well as climatic and regional context). Regarding the different facets of plant diversity, the three predictors were most tightly related to the trends in species richness and community-weighted mean niche width (with averages of 11.2% and 10.1% of explained variation, respectively) and least tightly related to the trends in functional divergence and functional richness (both with 4.8%). Regarding EUNIS level 1 habitat types, the three predictors were most tightly related to diversity trends in inland waters and vegetated man-made habitats (10.1% and 9.8%) and least tightly related to the diversity trends in grasslands and inland sparse vegetation (4.6% and 4.3%).

The specific annual percentage changes in local plant diversity among EUNIS level 1 habitats and habitat-change trajectories are shown in Fig. 5, Supplementary Figs. 7 and 8, statistics in Supplementary Data 6). Vegetation cover increased most strongly in inland waters, sparsely vegetated, and man-made habitats that underwent successional trajectories. Taxonomic diversity most strongly increased in mires and wetlands that underwent disturbance trajectories, potentially due to the establishment of functionally different species, as indicated by the coupled increases in functional richness. The number and cover of threatened Red List species increased most strongly in grassland and forest habitats that underwent successional trajectories and decreased (to a lesser degree) in stable shrublands and forests (Supplementary Fig. 8). Non-native species increased most strongly in mires and wetlands that underwent successional trajectories and decreased only (to a lesser degree) in stable sparsely vegetated and successional vegetated man-made habitats. With regard to the niche width indicators, specialist species decreased most strongly in marine and inland waters that underwent successional trajectories and increased most strongly in shrublands that underwent successional trajectories. Habitat generalists increased most strongly in mires and wetlands, especially with successional and disturbance trajectories, and decreased only in grasslands that underwent successional or disturbance trajectories. Jointly, this resulted in the largest positive

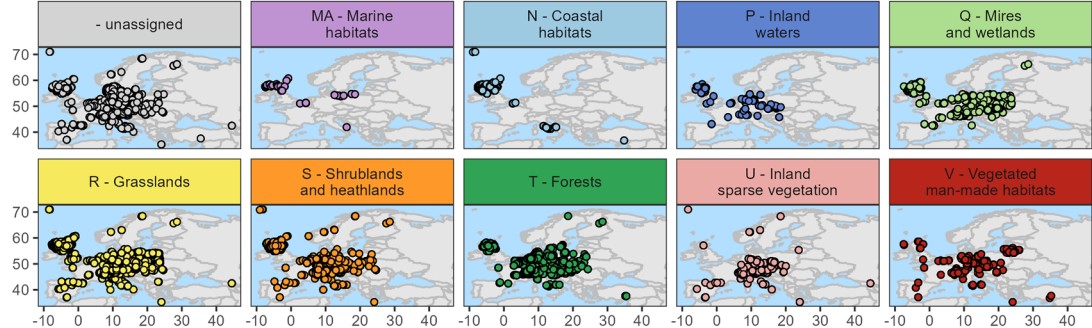

a) Distribution of time series

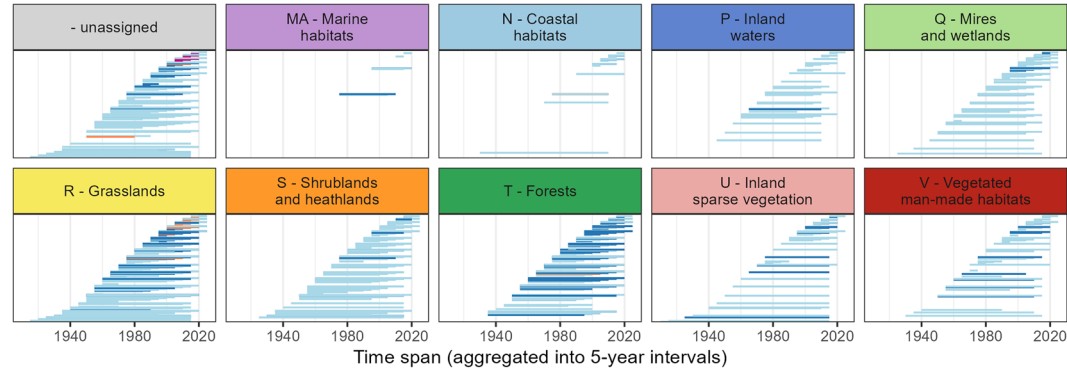

b) Temporal distribution of time series

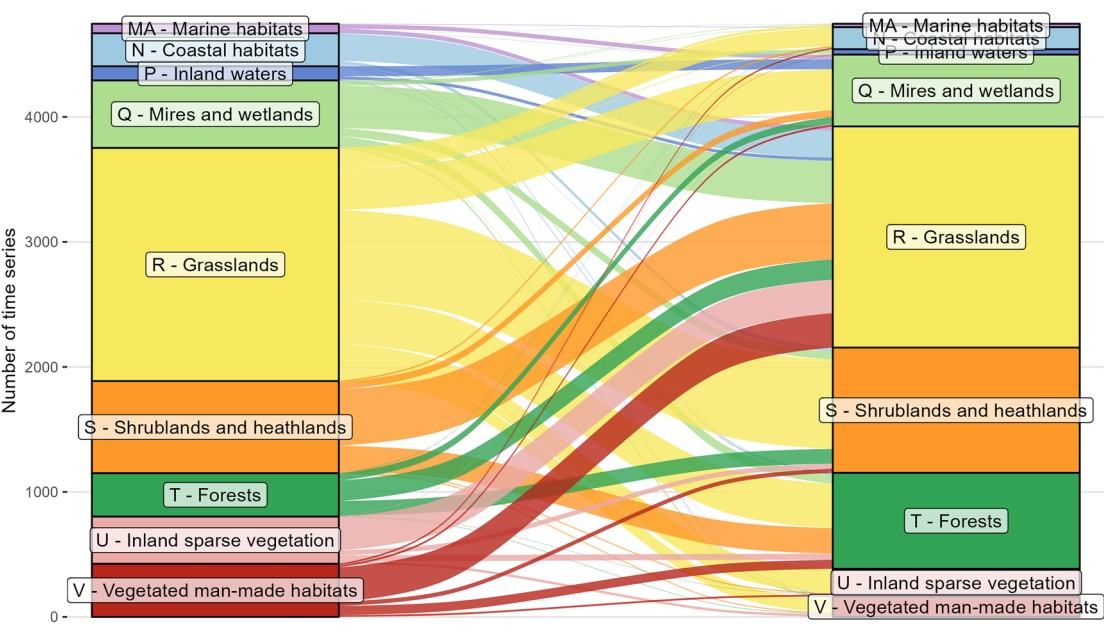

c) Changes in habitat types

**Fig. 3 | Data overview of the ReSurveyEurope time series of vegetation plots.**
**a** shows the geographic distribution of time series. **b** shows the temporal distribution of time series from the first to the last survey date, aggregated at 5-year intervals. **c** shows the number of time series with shifts in the assigned EUNIS level 1 habitat type between the initial (left) and the last survey date (indicated by the width of the connecting lines). Time series with stable habitat types and time series that could not be assigned to a habitat type were omitted from **c** (but are shown in Supplementary Figs. 5 and 6, and Supplementary Data 2). Maps were created with rnaturalearth[118,119].

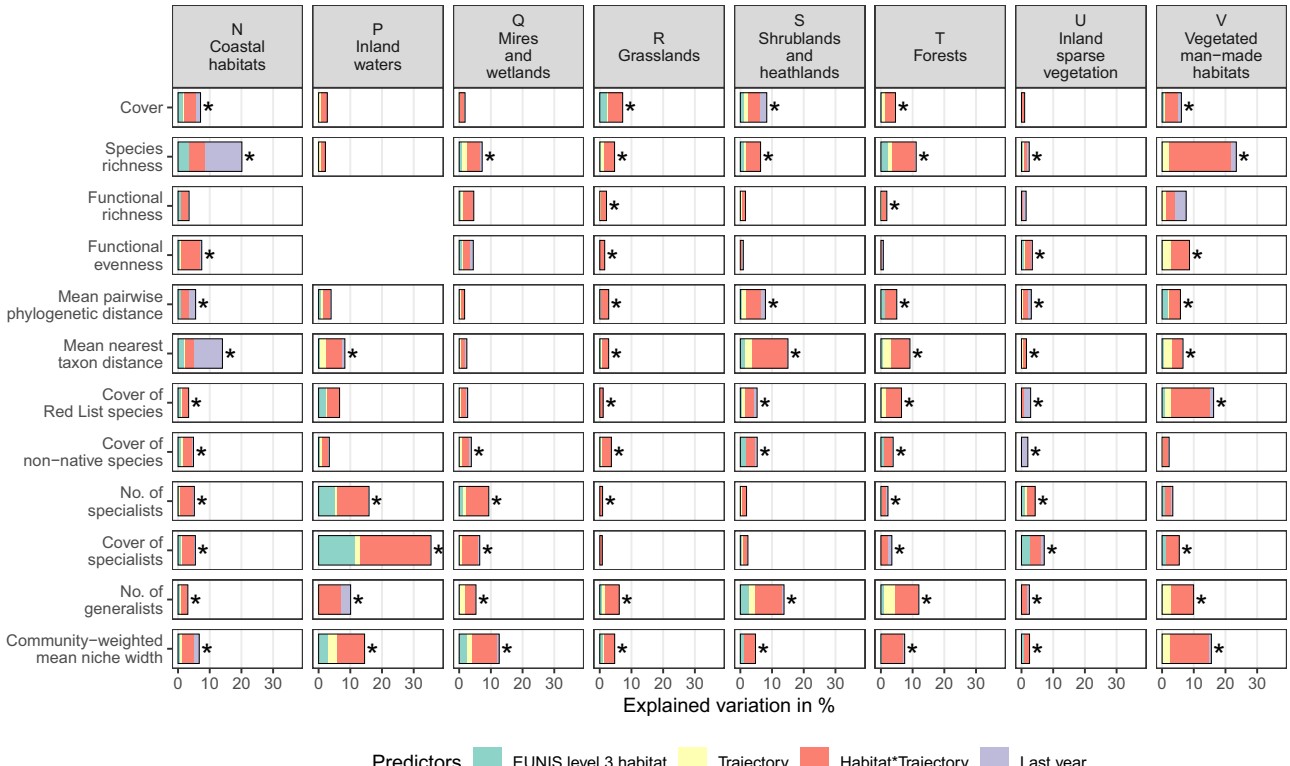

**Fig. 4 | Proportion of explained variation in annual percentage changes, relative to baseline conditions, for diversity indices of local plant communities, separated by EUNIS level 1 habitat types.** Bar lengths show the amount of partial explained variation, estimated with dominance analysis by EUNIS level 3 habitat type, habitat-change trajectory ("stable", "succession", or "disturbance"), their interaction, and the year of the last observation. Asterisks indicate significance of full models that included all four predictors (estimated with analyses of variance at $p < 0.05$, all statistics are shown in Supplementary Data 4). Analyses of functional diversity in P – Inland waters were excluded due to low sample sizes.

shifts in community-weighted mean niche width in marine, inland water, and grassland habitats that underwent successional trajectories (Supplementary Fig. 8).

## Trends in habitat-specific gamma diversity

In contrast to our hypothesis (H3), European gamma diversity, quantified as the summed number of unique taxa across all time series, did not follow a general or any temporal trends (Supplementary Fig. 9, statistics in Supplementary Data 7). Yet, only when differentiated among habitat types and habitat-change trajectories, we found a significant increase in European gamma diversity in stable grasslands and sparsely vegetated habitats that underwent successional trajectories (Fig. 6). Against our expectations, the absence of gamma diversity trends in nutrient-rich and disturbed habitats indicated a turnover instead of an invasion of non-native or habitat generalist species. Furthermore, the highest (albeit non-significant) proportion of negative gamma diversity trends in stable and successional forests might indicate large-scale losses of forest species that were not visible from local plot-level data alone. Even though our analyses revealed clear gaps in habitat types and time periods, they showed that European biodiversity changes involved a compositional re-organization rather than a uniform erosion of plant species richness.

## Vegetation time series for biodiversity monitoring

In this study, we synthesized the general and habitat-specific trends in multiple facets of local plant diversity from vegetation-plot time series that covered up to 113 years and the majority of European habitat types[33]. Contrary to recent findings[50,52,53], we found no consistent continental decline of plant diversity but, on average, an increase in local plant diversity that was associated with increases in non-native, threatened Red List, and habitat generalist species. Contributing to the

ongoing discussion on global versus local trends in plant diversity[8,16,54], we demonstrated that diversity trends in local plant communities defy broad generalizations as they reflect complex, habitat- and time-specific responses to disturbance and successional dynamics[26].

Discrepancies with recently published net-zero[52] or deteriorating[50,53] trends for European local plant diversity highlight the added insights from repeated vegetation observations (compared to one-time sampled vegetation plots[52]) over large ecological gradients (compared to ecologically limited time series[50,53]). Yet, in agreements with extensive vegetation-plot time series across the Czech Republic and Germany[16,32], we conclude that the observed increases in local species richness are not necessarily associated with increases in habitat quality, but tend to be driven, at least in part, by increases in non-native and generalist species—both of which could mask ongoing declines in native or specialized species[15,16,55–59].

The observed variation in plant diversity trends among habitat types[16,52,60,61] reflect the impacts of varying ecological drivers. For instance, positive diversity trends in stable grassland habitats might partly be attributed to upward shifts of generalist and non-native plants that follow shifting isotherms within mountain ranges[62–66]. Increases in cover and diversity of mire and wetland communities might be unrelated to conservation efforts, but rather reflect general reductions in habitat quality from desiccation and disturbances that allow the establishment of shrubs, trees, and more generalist grass species[63,67–70] (Supplementary Figs. 7 and 8). Here, future resurveys are needed to evaluate whether reported increases in plant diversity could finally lead to a general decline of plant diversity in these peculiar habitats. Finally, increases in vegetation cover and threatened Red List species in man-made habitats could be partly caused by land-use extensification and abandonment, both of which might lead to higher conservation value of these artificial habitats[71–73].

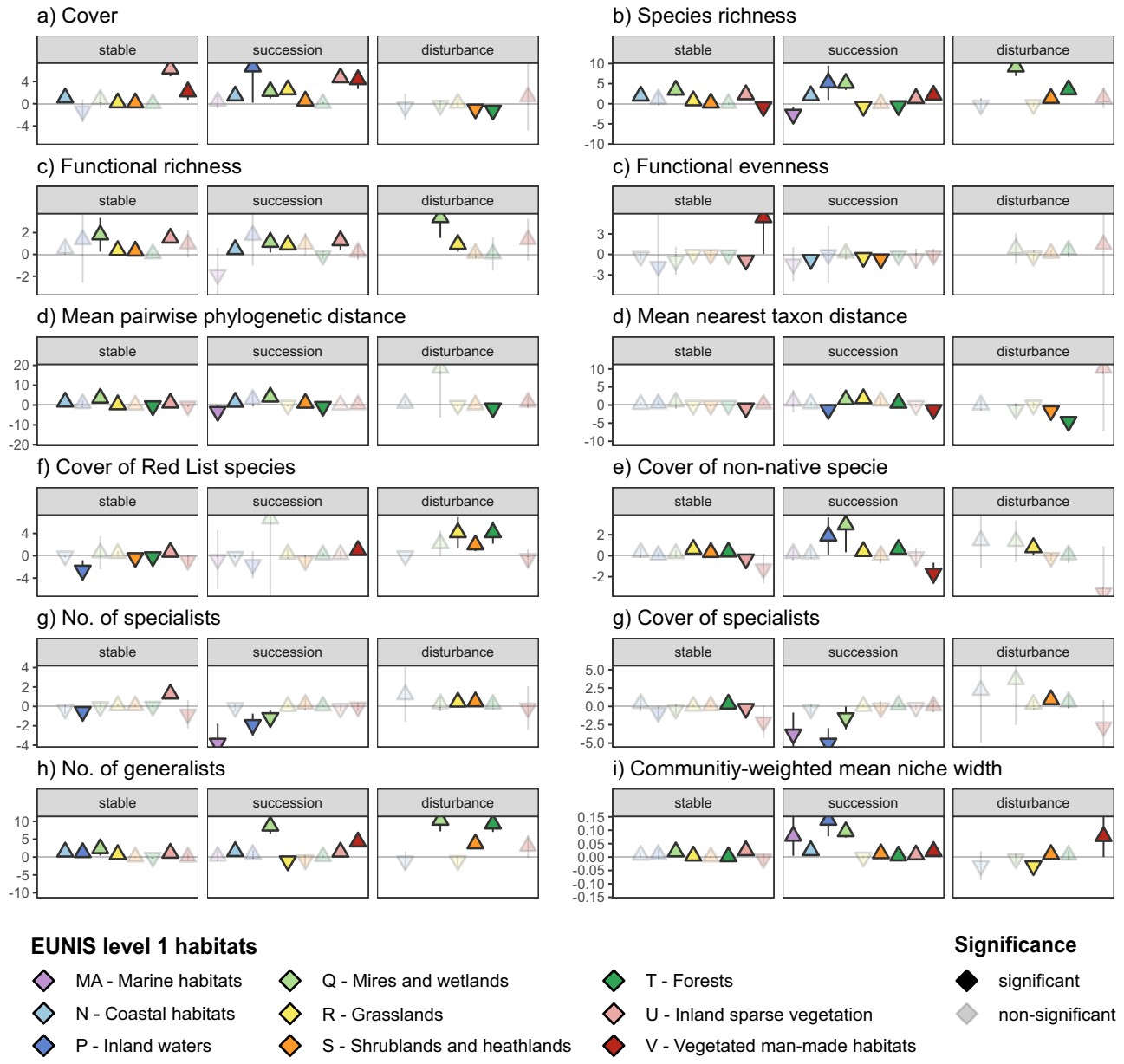

**Fig. 5 | Average annual percentage changes in diversity indices (relative to baseline conditions) in local local plant communities of different habitat types and habitat-change trajectories.** The different diversity indices in panels **a**–**i** are described in Fig. 1. Subgroup-specific average percentage changes were calculated with weighted linear models and their significance was tested with separate two-sided Student's *t* tests (at *p* < 0.05). Error bars show Wald-approximated 95% confidence intervals, calculated as weighted average ± 1.96 * standard error.

To better understand the drivers and ecological consequences of the observed species losses and gains, more studies on temporal species turnover at finer spatial scales are needed[74–76]. Although our results are derived from a comprehensive dataset of European vegetation-plot time series, we cannot rule out that the observed diversity trends were, at least in part, affected by methodological artifacts, such as pseudo-turnover[77], and anticipated species occurrences from historic records. As most of the vegetation-plot time series compiled in ReSurvey Europe[33] were initially established to address specific ecological questions, it is important to acknowledge that the synthesized diversity trends may, at least in part, be biased by variation in resurvey design (c.f., Supplementary Table 6), different plot sizes (c.f., Supplementary Fig. 10, statistics in Supplementary Data 8), and temporal changes research priorities, species taxonomies, and sampling effort[78]. For instance, disturbance trajectories are likely underrepresented because historic vegetation plots that have been converted to artificial or agricultural land are less frequently resurveyed. Likewise, time series from aquatic and coastal habitats remain both underrepresented and spatially clustered, as do resurveys of existing vegetation plots on the Iberian Peninsula, in Fennoscandia, and in the easternmost regions of Europe. Although some of the imbalances could be alleviated through rarefaction and spatially weighted analyses, closing of these gaps ultimately requires future resurveys of existing vegetation plots by botanical experts (motivate-biodiversity.eu), potentially complemented by AI-assisted citizen-science approaches, such as Observation.org, iNaturalist (www.inaturalist.org), Pl@ntNet (plantnet.org), or Pl@ntBERT[79] (github.com/cesar-leblanc/plantbert). To promote resurveys of established vegetation plots, the locations of nearly all plots in the European Vegetation Archive and the ReSurveyEurope database have now been made openly accessible at www.evamap.eu[80].

In this study, we demonstrated how a massive dataset of vegetation-plot time series can be leveraged to improve our

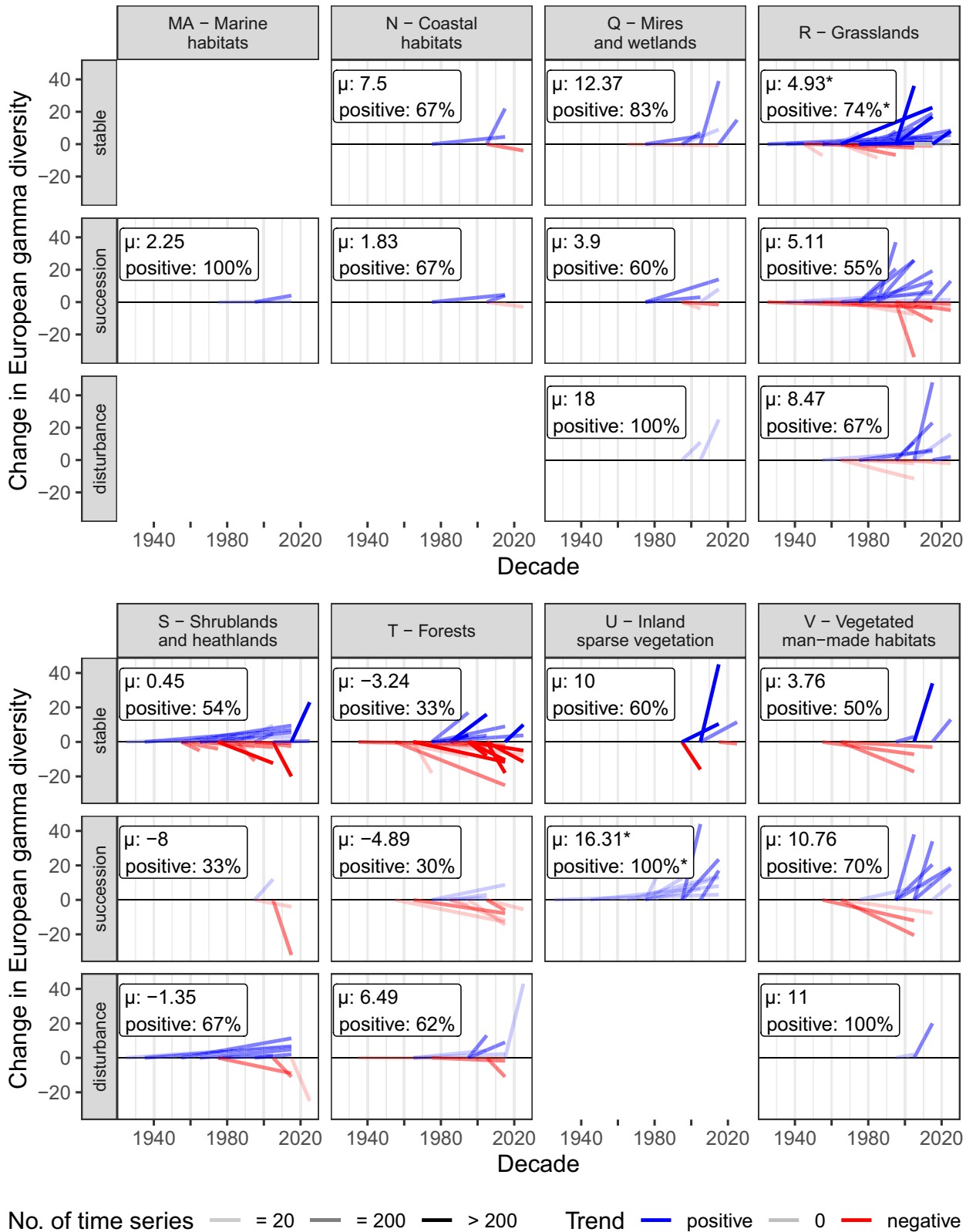

**Fig. 6 | Decadal trends in gamma diversity separated by EUNIS level 1 habitat type and habitat-change trajectory.** To account for temporal trends in sample sizes, each trend line includes only pairs of observations from vegetation plots that were conducted in the first and the second decade compared. In the boxes, the first number shows the average decadal trend in gamma diversity (with asterisks indicating significance according to two-sided Student's t tests at $p < 0.05$) and the second number shows the proportion of positive versus negative decadal trends (with asterisks indicating significance according to two-sided binomial tests at $p < 0.05$). Statistics are shown in Supplementary Data 7.

understanding of large-scale and specific trends in local plant diversity that are not (yet) accessible through satellite-derived remote sensing products[81]. Yet, the compiled dataset of vegetation-plot observations could serve as ground-truthing data or provide plant functional type classifications for future regional- to large-scale projections of plant diversity and ecosystem functioning trends. To improve upon the relatively modest explanatory power of up to 24.9% of variation in local diversity trends, we recommend that future syntheses should account for additional factors that could lead to varying trends in local plant diversity. First, historical datasets should be checked for methodological inconsistencies, including differences in sampling purposes, spatial scales, and unbalanced gradients in environmental conditions or community composition[82]. Secondly, future studies should explicitly account for variation in key drivers of local biodiversity change, such as changes in land-use intensity—also in the surrounding landscape[29,83]—as well as changes in abiotic conditions, eutrophication levels, or the protection status of resurveyed plant communities[64,84–86].

Synthesized over a century of European vegetation-plot time series, we detected prevailing positive trends in the diversity of local plant communities. However, a substantial proportion of these positive trends might be rather related to increases in non-native and habitat generalist species, both of which might have masked non-significant or declining trends among habitat specialists. For example, in mire and wetland communities, we observed a clear decline in the diversity of habitat specialists. Yet, positive trends in threatened Red List species and habitat specialists after the year 2000 also show that biodiversity recovery is possible under favourable management[52]. Still, due to the complexity of ecological systems, also with regard to the intricate effects of climate and land-use change, the site- and habitat-specific future trends in local plant diversity cannot yet be reliably extrapolated from the compiled vegetation-plot time series. Rather, our results emphasize that future syntheses and modelling approaches should account for habitat types and anticipated habitat-change trajectories to generate more accurate predictions for the impacts of different management options to foster the conservation of local plant diversity.

## Methods

### Vegetation data
Raw vegetation data consisted of 429,917 vegetation-plot observations from 119,964 permanent and semi-permanent vegetation plots; the latter ones were not permanently marked but resampled in proximate and representative plant communities. Time series of vegetation-plot data were collected and curated by the ReSurveyEurope initiative[33] and accessed via the European Vegetation Archive[87] on April 4th, 2024 (project no. 200, databases listed in Supplementary Data 9). Vegetation-plot data were exported as species-level percentage cover values of vascular plant species in one or multiple vegetation layers, depending on the raw data submitted to ReSurveyEurope. To guarantee consistent analyses over homogeneous data structures, we excluded plots that were experimentally manipulated for research purposes but kept all plots that were regularly mown or managed following traditional or regular management practices. We excluded all vegetation-plot observations that consisted only of species presence/absence data without species cover information because this data did not permit the assignment of EUNIS habitat types or the calculation of most indices of plant diversity. We further excluded all time series in which the area of the resurveyed vegetation plot changed by a factor of more than two because biodiversity patterns and trends are area-dependent[88,89]. The filtered dataset included 199,282 vegetation-plot observations belonging to 57,390 different time series.

### Assignment of EUNIS habitat types and habitat-change trajectories
Using information on species abundances and biogeographic regions, we applied an automated, expert-based classification system that

assigns vegetation-plot observations to the EUNIS hierarchy of habitat types[90,91]. In total, we were able to assign 107,040 vegetation-plot observations to broadly defined EUNIS level 1 habitat types (53.7%), of which we could classify a subset of 69,389 vegetation-plot observations to more narrowly defined EUNIS level 3 habitat types (34.8%). Vegetation-plot observations that could be assigned to multiple EUNIS habitat types or consisted of multiple observations were assigned to the matching level 1 or level 2 habitat types, e.g., when older plots could not be exactly re-located so that the re-sampling was conducted in proximate representative habitats and with sometimes differing plot numbers so that diversity trends had to be estimated in these so-called N-to-N relationships of semi-permanent plots. Based on the assigned EUNIS level 1 and level 3 habitat types during the initial versus the last observation date, we classified each time series into the following classes of habitat-change trajectories (see Supplementary Data 3): stable time series that remained at the same or similar EUNIS level 3 habitat type, successional time series that were characterized by a change in EUNIS habitat type that implied an increase in biomass and/or habitat complexity, disturbed time series that were characterized by decreasing biomass and/or habitat complexity due to natural or human-caused disturbances, other time series for which the observed shifts in habitat types indicated a shift in land-use or could not be categorized to the previous three trajectories (e.g., shifts from wet to dry heath, or shifts from dark taiga to Picea mire forest), and unclassifiable time series that included all time series for which the initial and/or last observation could not be assigned to a EUNIS level 3 habitat type.

### Taxonomic harmonization
Within each time series, we checked for potential name mismatches, which we defined as taxa observations that had different names or were split into multiple taxa in subsequent vegetation-plot observations but actually referred to the same taxon, e.g., due to different surveyors, uncertainties in taxon determination, or altered taxonomies over time. For this, we compiled all instances where a taxon was lost or gained between two consecutive vegetation-plot observations while other taxa of the same genus were still present. In search for systematic name mismatches, we extracted all losses and gains that occurred in >40% of a minimum of five vegetation plots located in the same research site of a given resurvey project. After consultation with the original data owners, we hand-corrected 24 project- and site-specific taxon names, totalling 3097 occurrence observations (Supplementary Table 7). To be able to link our taxon names with those of the TRY Plant Trait Database[39], we matched all taxa to the taxonomic backbone of the sPlot – Global Vegetation Database, version 4.0[92] and hand-corrected 2944 taxon names that could either be matched to synonyms in the sPlot taxonomic backbone, were flagged as synonyms according to the World Flora Online Database (version 2023.12,[93]), or had identifiable spelling mistakes. The obtained matched/corrected taxa were unified to the species or genus level (when species information was not available), resulting in a total of 4963 taxon names with 2,836,096 occurrence observations.

### Species-level information
Species functional traits were extracted from a gap-filled dataset that covered 46 mean trait values (Supplementary Table 1) for 78,278 taxa (including genus-level trait means). These taxon-level mean values were calculated from the raw data of the TRY Plant Trait Database, version 6.0[39], using Bayesian Hierarchical Probabilistic Matrix Factorization to predict missing values from observed trait records and phylogenetic relationships[94]. For our analyses, we log-transformed and standardized all trait values to have a mean of zero and a variance of one, and extracted the subset of 3834 taxa that occurred in our time series of vegetation plots. To reduce multicollinearity among traits, we visually inspected correlation plots (Supplementary Fig. 11) and the

first eight principal components that we calculated for the species-by-trait matrix (with 3834 species rows and 46 traits columns, Supplementary Fig. 12). Based on pairwise-correlations of Pearsons's $r > \pm 0.6$, we decided to only keep the following 14 log-transformed and standardized traits for all statistical analyses: plant height and stem diameter (as a proxy for stature); rooting depth, specific root length, and specific length of fine roots (root architecture); leaf carbon-to-nitrogen ratio, leaf phosphorus content, and leaf dry matter content (leaf physiology); leaf area, leaf thickness, and specific leaf area (leaf architecture); stem conduit density and stem conduit diameter (hydraulic strategy); and seed mass (as a proxy for germination and seedling strategy).

Species phylogenetic relationships were extracted from the phylogenetic backbone of the sPlot database, version 3.0[20,95]—a super tree based on the Open Tree of Life[40] with 589 additional species on 634 nodes that sum up to 7857 species. Taxa that were not resolved by the phylogenies of origin were bound to the most recent common ancestor if the genus included more than one species, or to half of the terminal level of a sister species if only one species was available in the focal genus. The obtained phylogenetic backbone for our time series of vegetation plots covered 4684 taxa (95.1%, see Supplementary Data 2).

Species threat status was extracted in binary form ("threatened" versus "not threatened") from a harmonized, static compilation of the European Red List and the National Red List Project (www.nationalredlist.org[96,97]). We classified species as "threatened" when they were listed as Critically Endangered (CR), Endangered (EN), Vulnerable (VU), Possibly Extinct (PE), Possibly Extinct in the Wild (PEW), or other similar threat levels that were difficult to match to an IUCN category (e.g., Rare, Nationally Critical, Sparse, Declining, etc.), but were clearly indicative of a "threatened" status. We classified species as "not threatened" when they were either not listed in the Red Lists or listed as Near Threatened (NT) or Least Concern (LC), according to the IUCN classifications. For our analyses, we classified species as Red List species if they were classified as "threatened" on either the European Red List or on the national Red Lists of the country in which the vegetation plot was situated.

For each species, the origin in Europe was assessed following delimitation in the FloraVeg.EU database[42]. All plant species that are native to at least part of Europe were considered native. Species introduced intentionally or unintentionally by humans to Europe from other continents were considered non-native plants.

Species-specific relative niche width was calculated as a proxy for habitat specialization. A co-occurrence-based approach from Fridley[43] was applied following the basic assumption that habitat generalists should co-occur with a larger number of species than specialist species that are restricted to a limited set of habitats and co-occurring species. To obtain representative estimates, we restricted all niche width calculations to species with ≥50 plot occurrences ( = 1995 species). For each species, we randomly selected a subset of 20 vegetation-plot observations from different time series and calculated the multiple Simpson dissimilarity index[45], which is independent of species richness and nestedness, meaning that plant communities with fewer but the same species are considered to be compositionally similar. To obtain representative estimates, this random selection and calculation were repeated 100 times for each species, and the niche width was then quantified as the mean value of the resulting 100 multiple Simpson dissimilarity values. The obtained species-specific niche width estimates ranged from 0.7 to 0.98 (Supplementary Fig. 13), and we classified all species with the lowest 10% of values as habitat-specialists and all species with the highest 10% of values as habitat-generalists.

## Community-level diversity indices

As species could be recorded in different vegetation layers of the same vegetation plot, we first calculated the summed cover for each species across all layers, using equation six from Fischer[98]—shown in Fig. 1. With these taxon-level cover values, we calculated several community-level indices of plant diversity (Fig. 1). Vegetation cover was quantified as the proportion of plot area that was covered by vegetation (under consideration of overlaps among co-occurring taxa[98]). Taxonomic diversity was quantified by the number and evenness in cover of co-occurring taxa. Functional diversity was quantified as functional richness, functional evenness, and functional divergence of the selected 14 traits (all normalized to unit variance[99–103]). Since these functional diversity indices are known to be sensitive to the completeness of the available trait data[104], we only calculated them for vegetation-plot observations for which the summed cover of all plants with trait data available was ≥80% of the total vegetation cover. Phylogenetic diversity was quantified as the sum of branch lengths from phylogenetic trees that were pruned to the level of vegetation-plot observations (Faith's phylogenetic diversity), mean pairwise phylogenetic distance, and mean nearest taxon distance. Phylogenetic diversity indices were only calculated for those vegetation-plot observations for which we had the phylogenetic relationships for all co-occurring taxa in the respective observation. For the subset of "threatened", "specialist", and "generalist" species, we calculated the summed cover and species richness using the same methods applied to the entire plant community. To account for shifts in species with "intermediate" niche width, we supplemented the "specialist" and "generalist" values with the community-weighted mean niche width, which we calculated from species' relative cover and only for vegetation-plot observations for which we had niche width estimates for species that together covered ≥80% of total plant cover.

## Time series-specific trends in diversity indices

For each time series, i.e., for each set of vegetation-plot observations, we used separate ordinary least square regressions for each biodiversity index to calculate (1) the time series-specific annual percentage change and (2) the time series-specific annual change in the original unit of each biodiversity index. Time series-specific annual percentage changes were quantified by regression slopes between the observation year and the logarithm of the focal diversity index—to which we added a constant value of 0.5 in the case of the number of Red List, non-native, specialist, and generalist species. The obtained regression slopes were transformed to percentage values with $(e^{slope} - 1){*}100$. Time series-specific annual changes in original units of the focal diversity indices were quantified by regression slopes between the observation year and the original values of the focal diversity index.

For time series with only two vegetation-plot observations (i.e., a baseline observation and a single resurvey), the obtained regression slopes quantified the time series-specific relative and absolute change in diversity values, divided by the timespan between the two observations (in years). For time series with more observations, the resulting regression slopes quantified the time series-specific relative and absolute annual change in the biodiversity value. Time series-specific diversity trends could cover multiple vegetation plots, e.g., in the case of N-to-N relationships from semi-permanent resurvey plots for which one or more original vegetation plots were resurveyed with several new plots in proximate locations in the same habitat instead of specific plot locations[33].

## Average trends in local plant diversity

For each diversity index, we used linear models to estimate two metrics of weighted biodiversity trends. (1) Average annual percentage changes synthesized diversity trends across all time series; each one contributing according to the logarithm of the number of observations (to upweight the impact of time series with many observations). (2) Balanced annual percentage changes synthesized diversity trends across all EUNIS level 3 habitat types; each time series being assigned with a weight that guaranteed equal impact of each habitat type. For

this analysis, we excluded all time series from EUNIS level 3 habitat types with fewer than ten time series in our dataset. Significance of weighted biodiversity trends was determined from Wald-approximated 95% confidence intervals (estimate ± 1.96 × standard error).

We conducted additional sensitivity analyses to test how the average and balanced annual percentage changes could have been affected by differences in resurvey design. For this, we re-calculated the weighted linear models for the average and balanced trends of each diversity index—but added the following two variables as fixed effects: (1) the type of resurvey design (semi-permanent versus permanent) and (2) the time series-specific relative change in plot size between the baseline and the last resurvey observation.

To test for differences in biodiversity trends before and after the year 2000, we separated all vegetation-plot observations into two groups (also within individual time series): (1) observations conducted before 2000 and (2) observations conducted after 2000. For each group, we re-calculated and tested the significance of average annual percentage changes analogue to the complete dataset. Differences in average annual percentage changes between the two groups were tested with weighted $t$-test with an analogue weighting scheme, i.e., with each time series-specific trend being weighted according to the logarithm of the number of observations. Due to data limitations, these comparisons between pre- versus post-2000 time series could only be conducted for the subset of time series that were classified as EUNIS level 1 grassland, shrubland, or forest habitats.

### Specificity of local biodiversity trends

Separately for each combination of EUNIS level 1 habitat type and biodiversity index, we applied linear model-based dominance analyses[105,106] to calculate the predictive power of the following four variables to explain the observed variation in annual percentage changes: the type of EUNIS level 3 habitat, the type of habitat-change trajectory ("stable", "succession", or "disturbance"), the interaction among habitat type and habitat-change trajectory, and the year of the last observation. To avoid bias from low sample sizes, we restricted all dominance analyses to those combinations of EUNIS level 1 habitat type and biodiversity index that had ≥2 different EUNIS level 3 habitat types, each with ≥2 different habitat-change trajectories, each with ≥10 time series (i.e., a minimum of $2 \times 2 \times 10 = 40$ time series).

For each combination of diversity index, EUNIS level 1 habitat type, and habitat-change trajectory, we applied linear regression to estimate the specific average annual percentage change; with each time series contributed according to the logarithm of the number of observations (to upweight the impact of time series with many observations). Significance of weighted biodiversity trends was (analogue to average diversity trends) determined from Wald-approximated 95% confidence intervals.

### Trends in European gamma diversity

To quantify the trends in European gamma diversity (i.e., in the cumulative number of unique taxa per EUNIS level 1 habitat type and habitat-change trajectory), we had to make sure that our analyses were not biased by different numbers of vegetation-plot observations at different time points. This might occur as a result of different numbers of time series in a given time period or N-to-N relationships (see above). To achieve this, we calculated the differences in gamma diversity between all possible pairs of decades (e.g., 1920–1929 versus 1930–1939). For each pair of decades, we selected those vegetation plots that had observations in both decades. When a time series had more than one observation in the same decade, we kept only the first observation in the earlier decade and the last observation in the later decade. For each pair of decades, we calculated the difference in gamma diversity, i.e., the difference in the number of uniquely recorded taxa across all vegetation-plot observations, divided by the temporal distance (in decades). Across all decadal trends in gamma

diversity, we determined the significance of the average gamma diversity trends with a $t$ test (at $p < 0.05$), and we determined the significance of positive versus negative gamma diversity trends with a binomial test (at $p < 0.05$).

### Software

All analyses were conducted in R[107] using the following packages: dominance analyses were calculated with dominanceanalysis[108], functional diversity and phylogenetic diversity indices were calculated with FD and picante[103,109,110], graphical representations were plotted with data.tree, ggplot2, ggalluvial, ggh4x, and networkD3[111–116], linear mixed-effects models were calculated with lme4[117], map data were retrieved with rnaturalearth[118,119], principal component analyses were calculated with FactoMineR[120], weighted $t$-tests were calculated with weights[121], and the World Flora Online database was handled with WorldFlora[93,122].

### Reporting summary

Further information on research design is available in the Nature Portfolio Reporting Summary linked to this article.

## Data availability

The data generated in this study, i.e. time series-specific linear trends in all biodiversity indices, expressed in annual percentage changes and in absolute units, are openly available at the data repository of the German Centre for Integrated Biodiversity Research (iDiv) Halle-Jena-Leipzig (idata.idiv.de/ddm/Data/ShowData/3611, https://doi.org/10.25829/idiv.3611-rgwa69). Raw vegetation data cannot be made openly available as they belong to the owners and custodians of each vegetation database—but can be requested at the European Vegetation Archive[123]. Individual vegetation databases for this study are listed in Supplementary Data 9. Plant trait data can be downloaded from the website of the TRY plant trait database (www.try-db.org). Phylogenetic data can be downloaded from the website of the Open Tree of Life (tree.opentreeoflife.org). Classification into native and non-native species is openly available at the FloraVeg.EU database[42]. Red List data was downloaded from github.com/istaude/european-redlist-synthesis. A harmonized version can be requested from Laura Méndez (laura.mendez@ufz.de).

## Code availability

The R-code generated to harmonize taxa names, filter vegetation-time series data, calculate biodiversity indices, and analyse the general and specific trends in local plant diversity is openly available at github.com/StephanKambach/Local_trends_in_plant_diversity (zenodo.org, https://doi.org/10.5281/zenodo.18875713).

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

## Acknowledgements

The MOTIVATE project (motivate-biodiversity.eu) was funded by Biodiversa + , the European Biodiversity Partnership, under the 2022-2023 BiodivMon joint call. It was co-funded by the European Commission (GA No. 101052342) and the following national funding organisations: Austrian Science Fund FWF (MOTIVATE, pr.no. I 6846-B), Deutsche Forschungsgemeinschaft (DFG) project number 532411638, Research Council of Finland (RCF) grant number 359866, Spanish PCI2023-2 International Collaboration Projects number MCINN-24-PCI2023-146014-2, Technology Agency of the Czech Republic (TAČR) project number SS73020008, and Italian Ministry of University and Research (MUR) project number BIODIV22_00086. Views and opinions expressed are those of the authors only and do not necessarily reflect those of the European Union or the European Research Council. Neither the European Union nor the granting authority can be held responsible for them. We acknowledge support from the "VegCHange" project within the National Research Program 82 "Biodiversity and Ecosystem Services" of the Swiss National Science Foundation for J.De. (SNSF; grant No. 408240_235006) and funding by the European Union for K.V.M. (ERC starting grant FutureNature No. 101076837). Illustrations were designed with free templates from macrovector/Freepik. We thank all the researchers who collected valuable vegetation-plot data in the field and made them available in the ReSurveyEurope database.

## Author contributions

S.K., U.J., A.T.R.A., J.M.A.-M., M.B., M.Ca., M.Ch., F.E., K.F., M.G., G.J.A.H., T.H., F.J., B.J.-A., I.K., J.L., R.N., F.M.S., A.V., and H.B. developed the research idea. U.J., A.T.R.A, E.B., I.B., M.B.-R., I.B, G.B., M.L.C., A.Ch., M.Ch., J.De., M.D.S., J.Di., J.Do., S.D., F.E., K.F., E.G., B.G., M.H., E.I., F.J., A.J., B.J.-A., I.K., J.L., B.L., F.N., R.J.Pa., P.Pe., R.P., V.R., F.M.S., W.S., G.S., L.T., M.T., K.V.M., V.V., K.V., M.V., T.W., H.B., and many more partners of ReSurveyEurope contributed vegetation data (euroveg.org/resurvey). I.K., S.K., G.D., and A.V. curated the vegetation data. A.T.R.A., H.B., B.E.L.C., M.B.C., V.F., A.G.-R., S.J., W.D.K., G.L., U.L., F.L., A.P.M., J.M., A.S.M., A.N., R.O., J.P., B.X.P., P.Po., C.Rosch., C.Ross., B.S., D.S.C., S.S., T.A.S., M.J.S., N.G.S., R.T., E.W, and many more partners contributed plant trait data. L.M. curated the Red List Data. G.J.A.H. curated the species phylogenies. I.A. curated the classification into non-native species. M.Ch. assigned vegetation-plot observations to EUNIS habitat types. U.J. coordinated the MOTIVATE project. U.J., H.B., F.J, F.E., B.J.-A., F.M.S., M.Ca., M.Ch., and R.N. acquired funding. S.K. conducted the analyses, created the figures and wrote the manuscript with substantial contributions from U.J. and H.B., and contributions from all co-authors.

## Funding

## Competing interests

The authors declare no competing interests.

## Additional information

Stephan Kambach [1] ✉, Ute Jandt [1,2], Alicia Teresa Rosario Acosta[3], Jose Manuel Álvarez-Martínez [4,5], Irena Axmanová [6], Manuele Bazzichetto[7], Erwin Bergmeier [8], Markus Bernhardt-Römermann [2,9,10], Idoia Biurrun [11], Gianmaria Bonari [12,13], Marta Carboni[3], Marcos Bergmann Carlucci [14], Maria Laura Carranza [13,15], Bruno Enrico Leone Cerabolini [16], Alessandro Chiarucci [7], Milan Chytrý [6], Gabriella Damasceno [1,2], Jürgen Dengler [17,18], Michele De Sanctis [19], Jan Divíšek [6], Jiří Dolezal [20,21], Stefan Dullinger [22], Franz Essl [22], Klára Friesová [6], Veronika Fontana[23], Emmanuel Garbolino [24], Michael Glaser [22], Ana González-Robles [25], Behlül Güler[26], Georg J. A. Hähn [7], Michal Hájek[6], Tracy Hruska [27], Estela Illa [28,29], Florian Jansen[30], Steven Jansen [31], Anke Jentsch [18,32], Borja Jiménez-Alfaro[4,5], W. Daniel Kissling [33], Ilona Knollová[6], Gianalberto Losapio[34], Udayangani Liu [35], Jonathan Lenoir [36], Frederic Lens [37,38], Bernd Lenzner [22], Antonio J. Perea [39], Laura Méndez [2,40], Julie Messier [41], Akira S. Mori [42], Francesca Napoleone [19], Roger Norum[27], Alexander Novakovskiy [43], Renske Onstein [37,38], Robin J. Pakeman [44], Josep Peñuelas [45,46], Petr Petřík [20,47], Remigiusz Pielech [48], Bruno X. Pinho [49], Peter Poschlod[50], Valerijus Rašomavičius [51], Christiane Roscher [2,40,52], Christian Rossi [53], Francesco Maria Sabatini [7], Brody Sandel[54], David Schellenberger Costa [2,55], Wolfgang Schmidt [56], Serge Sheremetiev [43], Tanvir Ahmed Shovon[57], Marko J. Spasojevic[58], Nathan G. Swenson [59], Grzegorz Swacha [60], Rubén Tarifa [25,61], Lubomír Tichý[6], Marcello Tomaselli[62], Alicia Valdés[4,5], Koenraad Van Meerbeek [63,64], Vigdis Vandvik [65], Kiril Vassilev [66], Martin Večeřa [6], Evan Weiher [67], Thomas Wohlgemuth[68] & Helge Bruelheide [1,2]

[1]Institute of Biology/Geobotany and Botanical Garden, Martin Luther University Halle-Wittenberg, Halle, Germany. [2]German Centre for Integrative Biodiversity Research (iDiv), Halle-Jena-Leipzig, Leipzig, Germany. [3]Department of Science, University of Roma Tre, Rome, Italy. [4]Biodiversity Research Institute (IMIB), University of Oviedo–CSIC–Principality of Asturias, Mieres, Spain. [5]Department of Organismal and Systems Biology, University of Oviedo, Oviedo, Spain. [6]Department of Botany and Zoology, Faculty of Science, Masaryk University, Brno, Czech Republic. [7]BIOME Lab, Department of Biological, Geological & Environmental Sciences, University of Bologna, Bologna, Italy. [8]Department of Vegetation & Phytodiversity Analysis, University of Göttingen, Göttingen, Germany. [9]Institute of Biodiversity, Ecology and Evolution, Friedrich Schiller University, Jena, Germany. [10]Senckenberg Institute for Plant Form and Function Jena (SIP), Jena, Germany. [11]Department of Plant Biology and Ecology, University of the Basque Country UPV/EHU, Bilbao, Spain. [12]Department of Life Sciences, University of Siena, Siena, Italy. [13]NBFC, National Biodiversity Future Center (NBFC), Palermo, Italy. [14]Laboratório de Ecologia Funcional de Comunidades (LABEF), Departamento de Botânica, Universidade Federal do Paraná, Curitiba, Brazil. [15]EnvixLab – Department of Biosciences and Territory, University of Molise, Termoli, Italy. [16]Department of Biotechnology and Life Science (DBSV), University of Insubria, Varese, Italy. [17]Vegetation Ecology Research Group, Institute of Natural Resource Sciences (IUNR), Zurich University of Applied Sciences (ZHAW), Wädenswil, Switzerland. [18]Bayreuth Center of Ecology and Environmental Research (BayCEER), University of Bayreuth, Bayreuth, Germany. [19]Department of Environmental Biology, Sapienza University of Rome, Rome, Italy. [20]Institute of Botany of the Czech Academy of Sciences, Průhonice, Czech Republic. [21]Department of Botany, University of South Bohemia, České Budějovice, Czech Republic. [22]Department of Botany and Biodiversity Research, University of Vienna, Vienna, Austria. [23]Eurac Research Institute for Alpine Environment, Bolzano, Italy. [24]ISIGE - Institut Supérieur d'Ingénierie et de Gestion de l'Environnement, Mines Paris PSL, Fontainebleau, France. [25]Departamento de Biología Animal, Biología Vegetal y Ecología, Universidad de Jaén, Jaén, Spain. [26]Biology Education, Dokuz Eylul University, Izmir, Turkey. [27]Cultural Anthropology, University of Oulu, Oulu, Finland. [28]Department of Evolutionary Biology, Ecology and Environmental Sciences, University of Barcelona, Barcelona, Spain. [29]Biodiversity Research Institute (IRBio), University of Barcelona, Barcelona, Spain. [30]Landscape Ecology, University of Rostock, Rostock, Germany. [31]Institute of Botany, Ulm University, Ulm, Germany. [32]Disturbance Ecology and Vegetation Dynamics, University of Bayreuth, Bayreuth, Germany. [33]Institute for Biodiversity and Ecosystem Dynamics (IBED), University of Amsterdam, Amsterdam, The Netherlands. [34]Department of Biosciences, University of Milan, Milan, Italy. [35]Royal Botanic Gardens, Kew, UK. [36]UMR CNRS 7058 "Ecologie et Dynamique des Systèmes Anthropisés" (EDYSAN), Université de Picardie Jules Verne, Amiens, France. [37]Naturalis Biodiversity Center, Leiden University, Leiden, The Netherlands. [38]Institute of Biology Leiden (IBL), Leiden University, Leiden, The Netherlands. [39]Department of Plant Biology and Ecology, University of Seville, Seville, Spain. [40]Department Community Ecology, Helmholtz Centre for Environmental Research (UFZ), Halle, Germany. [41]Department of Biology, University of Waterloo,

Waterloo, Canada. [42]Research Center for Advanced Science and Technology, University of Tokyo, Tokyo, Japan. [43]Institute of Biology of Komi Scientific Centre of the Ural Branch of the Russian Academy of Sciences, Syktyvkar, Russia. [44]James Hutton Institute, Cragiebuckler, Aberdeen, UK. [45]National Research Council (CSIC), Global Ecology Unit, CREAF-CSIC-UAB, Barcelona, Spain. [46]Center for Ecological Research and Forestry Applications (CREAF), Barcelona, Spain. [47]Department of Ecology, Czech University of Life Sciences, Prague, Czech Republic. [48]Institute of Botany, Jagiellonian University, Kraków, Poland. [49]Institute of Plant Sciences, University of Bern, Bern, Switzerland. [50]Ecology and Conservation Biology, University of Regensburg, Regensburg, Germany. [51]Nature Research Centre, Vilnius, Lithuania. [52]Department of Physiological Diversity, Helmholtz Centre for Environmental Research (UFZ), Leipzig, Germany. [53]Department of Geoinformation, Swiss National Park, Zernez, Switzerland. [54]Department of Biology, Santa Clara University, Santa Clara, CA, USA. [55]Systematic Botany and Functional Diversity Lab, Leipzig University, Leipzig, Germany. [56]Department of Silviculture and Forest Ecology of the Temperate Zones, University of Göttingen, Göttingen, Germany. [57]Department of Biology, University of Regina, Regina, Canada. [58]Department of Evolution, Ecology, and Organismal Biology, University of California Riverside, Riverside, CA, USA. [59]Department of Biological Sciences, University of Notre Dame, Notre Dame, IN, USA. [60]Botanical Garden, University of Wrocław, Wrocław, Poland. [61]Estación Experimental de Zonas Áridas (EEZA-CSIC), Almería, Spain. [62]Department of Chemistry, Life Sciences and Environmental Sustainability, University of Parma, Parma, Italy. [63]Department of Earth and Environmental Sciences (EES), KU Leuven, Leuven, Belgium. [64]KU Leuven Plant Institute, KU Leuven, Leuven, Belgium. [65]Department of Biological Sciences, University of Bergen, Bergen, Norway. [66]Institute of Biodiversity and Ecosystem Research, Bulgarian Academy of Sciences, Sofia, Bulgaria. [67]Department of Biology, University of Wisconsin - Eau Claire, Eau Claire, WI, USA. [68]Forest and Soil Ecology, Swiss Federal Institute for Forest, Snow and Landscape Research WSL, Birmensdorf, Switzerland. ✉e-mail: stephan.kambach@gmail.com

