## [Transparent Peer Review file · Nature Communications]

Habitat-specific trends in taxonomic, functional, and phylogenetic diversity in European plant communities over a century

Corresponding Author: Dr Stephan Kambach

Version 0:

Reviewer comments:

Reviewer #1

(Remarks to the Author)

This work is really impressive and has great potential but does not yet quite deliver on that potential.

The authors have used an impressive dataset of vegetation time-series in concert with plant species' functional trait and genetic information to explore trends in multiple facets of diversity over the last 100 years. They show that while species richness and phylogenetic increases at the site-level there is a concomitant decline in functional evenness. However, trends are highly influenced by other extraneous factors such as habitat type and trajectory over the time-series, indicating the nuances required to assess local biodiversity change.

The debate and controversy about the average trend in species-richness in local assemblages have been ongoing for over a decade. The lack of resolution reflects a combination of biased data and often slightly simplistic analyses. Resolving the question for such an ecologically vital and taxonomically broad group as plants, across such a broad domain as Europe, would be a major contribution on an important topic with broader societal relevance.

Irrespective of what the answer of such a reanalysis is, it would be wholly appropriate for publication in Nature Communications: it's such an ambitious and thoughtful analysis on such an important topic that I can certainly see myself - and many others - citing it for years to come. It is a very large and geographically extensive data set of plots, combined with trait and phylogenetic information; time spans are usually long enough to at least dilute the obscuring effects of short-term volatility, which plague interpretation of trends in short time series; the analyses try to not only elucidate the trends but to decompose variation among sites; and the analyses are explained clearly, making it easy to follow what was done. There is a lot to like! But, there are a few major concerns that mean I think publication of the manuscript as it is would be a mistake.

1. There is no sensitivity analysis to explore how far the patterns might have methodological rather than ecological explanations.

a: Plot area can vary by up to a factor of 2. While understanding that insisting on equal plot area might reduce the size of the data set, it would be naïve to expect a nearly twofold increase in plot area to have no effect on species richness.

Conservatively taking the exponent of the species-area relationship as 0.15, plot area alone could be causing a $\pm 10\%$ "change" in species richness between consecutive surveys. Do the results change substantially when only equal-area plots are analysed? Perhaps better, what is the empirical relationship between $\log(\text{Richness1}/\text{Richness2})$ and year when controlling for the effect of $\log(\text{Area1}/\text{Area2})$ in the model?

b: Some plots are permanent and others are semi-permanent. Do these show the same trend? Permanent plots are conceptually much purer. Effects having an ecological basis are expected to be stronger (larger coefficients) in the permanent plots, even if they are less significant because of reduced sample sizes.

2. The authors acknowledge that the data are not geographically representative but disappointingly make no effort to correct for the bias. The data are also probably not representative in terms of land-use trajectories. The reason these biases are important is that, without correcting for them, the paper simply can't estimate the true average trend in species richness in local assemblages in Europe. Without that, the set-up for the paper doesn't really work, and it's harder to see that it represents a major step forward in understanding - we're left with wondering whether the pattern we see is driven by haphazard sampling. The transition matrix between EUNIS habitats provides a way of weighting each time series, such that

those for under-represented 'transitions' (which might be sites that stayed in a particular, poorly sampled, habitat type) get upweighted relative to those for over-represented 'transitions'. This would provide the best available estimate of the actual average trend in each of the diversity metrics. Please add this change!

3. Species richness and Faith's PD are both modelled, which is great, but there's no parallel analysis of any functional richness measure, which is a shame as it would be lovely to be able to compare across all three dimensions. It might also give support to currently only marginally supported statements about change in functional diversity over time.

4. Not a request for more analysis, but the term "species pool" is used incorrectly throughout. Rather than being the set of species from which any of the communities in the data set might be assembled, the concept given this name in the manuscript is in fact gamma diversity for a particular habitat. Please fix this.

Minor issues (see also comments/suggestions in attached manuscript file):

- Captions and figures/boxes are generally not very easy to follow (though Fig 1 is nice); Box 1 places the symbols in the equations out of sequence relative to where they appear in the box itself, making it hard for the reader to join them to their equations. Figure 2's Sankey diagram is confusing (because it leaves out the sites that stayed the same - at least add the % of each habitat that do stay the same to the column at the left), and the transition lines don't align with the boxes they come from. Figure 5 would be better presenting the changes in percentage terms rather than absolute units that don't mean much to readers. Relating to this last point, the text would be improved by reporting the percentage changes too.
- It is claimed (line 275) that a majority of time series show an increase in species richness, but this is not demonstrated (not the same thing as the mean change being positive).
- Line 307-310: Ms suggests that increase in TD & FD in mires and wetlands could be caused by an increase in grass dominance - but wouldn't that reduce FD (and perhaps also TD)?
- The way time series are weighted (lines 718-721) is not correct.
- No real analysis of trends of generalist and specialist species, could perhaps be something easy to do to support conclusions made.
- Taxonomic harmonization to a list of species compiled recently will lead to an upward bias in species numbers over time. It would be much better to harmonize to an old list, so that entities newly recognised as species between surveys are treated consistently.
- The writing around threatened species needs to be a bit clearer. The list of which species are threatened is static, so having more threatened species in an assemblage is likely to be indicative of a recovery rather than a general decline. The authors clearly understand this but not all readers will.
- Lots of minor points indicated in the manuscript file.

(Remarks on code availability)

Reviewer #2

(Remarks to the Author)

(Remarks on code availability)

Reviewer #3

(Remarks to the Author)

Review Comments

The manuscript entitled "Habitat-specific trends in taxonomic, functional, and phylogenetic diversity in European plant communities during the last 100 years" by Kambach et al. provides a large-scale analysis of vegetation time series across Europe, quantifying general and habitat-specific trends in taxonomic, functional, and phylogenetic plant diversity, alongside changes in Red List species. The authors report a general increase in species diversity but contrasting response patterns across habitats. The manuscript stands out for the articulation and harmonization of vegetation time series derived from phytosociological surveys, which can synthesize the trajectory of local co-occurrence of plant species across Europe. This type of information cannot be obtained from satellite products or from the distribution of individual species alone, which makes the results of this manuscript particularly relevant. Overall, this is a high-quality contribution, but I have some concerns regarding a key point, which I summarize below. Additional minor comments are provided in the annotated text.

A. Main concerns regarding the hypotheses and their implications

(1) Replacement of specialists by generalists

The authors state that local plant communities show stable species richness but negative trends in functional and phylogenetic diversity due to the replacement of habitat-specialist "loser" species with more generalist "winner" species sharing similar ecological strategies and evolutionary histories (8, 38, 39).

It is not clear to me how the authors tested and demonstrated that the winners are indeed generalists with similar ecological strategies and evolutionary histories. Additionally, I assume that the specialist/generalist classification was applied only to native species; otherwise, this could create confusion, particularly in relation to hypothesis 3. If the winners are native

generalists, I would indeed expect similar ecological strategies and evolutionary histories, but if they are non-native species, I would expect the opposite pattern at least for the evolutionary history.

(2) Habitat types and habitat-change trajectories

The authors hypothesize that differences in local diversity trends are related to both habitat types and habitat-change trajectories, expecting the highest gains in sparsely-vegetated habitats undergoing succession and the highest losses in specialized wetland and forest habitats undergoing disturbance.

References would be useful here to support this statement. However, with regard to forest habitats, I do not fully agree with the authors' interpretation. Late-successional forests generally host fewer species than early-successional states due to the strong competitive "umbrella effect," which typically allows only a few shade-tolerant species to persist. High species richness in late-successional forests is usually restricted to tropical forests in large plots. Similarly, in grasslands there is evidence that increased biomass and light limitation are associated with richness declines (e.g. Borer et al., 2014, Nature). Thus, as a general hypothesis, one could state that successional trajectories may be associated with richness declines, while disturbance trajectories may be associated with richness increases.

More broadly, in this hypothesis the authors attempt to incorporate two factors, habitat type and habitat-change trajectory, which is also one of their main results. However, it is difficult to interpret these factors together and to integrate this hypothesis with Fig. 4 and the main text.

Finally, I am not sure if the authors are referring to particular forest and wetland types when writing "specialised wetland and forest habitats." If so, this should be specified, since no internal classification of habitats is provided in the manuscript.

(3) Expansion of the European species pool

The authors argue that species pools have increased over the last century due to the spread of non-native and generalist species, and hypothesize that man-made and disturbed habitats will show the largest increases, since these habitats are more susceptible to invasions (40, 41). I agree with this hypothesis. However, how can the reader distinguish whether the winners are native generalists or non-native species? This seems problematic, especially since the conclusions attribute the main changes to non-native species.

In addition, the hypotheses do not explicitly mention threatened species. Should we expect any particular patterns for them? Or is the assumption that threatened species are primarily specialists?

B. General observation

I found a partial but important disconnection between the hypotheses and the results, as well as a strong focus on richness patterns compared to the other response variables. The hypotheses raise strong expectations that are not fully met in the results and discussion. I encourage the authors to present their findings in a more comprehensive way that more directly addresses their hypotheses.

(Remarks on code availability)

Version 1:

Reviewer comments:

Reviewer #1

(Remarks to the Author)

I thank the authors for having engaged so thoroughly and thoughtfully with my comments and suggestions. This revised version of what was already a very impressive manuscript has allayed all my major concerns and also dealt entirely satisfactorily with all the minor points I raised. As I said in my initial review, I think this improved version absolutely meets the required novelty, ambition and rigour for publication in Nature Communications, and I very much look forward to citing it!

(Remarks on code availability)

Reviewer #2

(Remarks to the Author)

(Remarks on code availability)

Reviewer #3

(Remarks to the Author)

I really enjoyed reading the revised version of this manuscript and I appreciate the authors' detailed responses and the substantial improvements made. I have only one remaining comment.

In lines 232–235, the authors state that, because disturbed and nutrient-rich habitats tend to be more susceptible to invasive species, they expected time series from disturbed and man-made habitats to show comparatively larger increases in habitat-specific gamma diversity. However, this expectation is not explicitly revisited in the Results/Discussion (lines 345–356). As the analyses show largely non-significant responses across habitats and trajectories, and even negative trends in stable trajectories and successional forests, a deeper interpretation may indeed be too speculative. Nevertheless, it would be helpful to explicitly acknowledge that this expectation was not supported by the results. One possible explanation could be that recent European landscape trajectories have often been characterized by forest expansion and a reduction in disturbance intensity (e.g. grazing abandonment), which may limit the spread of non-native species. This would contrast with patterns reported from regions experiencing forest degradation and rapid urban expansion, where increases in non-native species are more commonly observed.

(Remarks on code availability)

Reviewers #1 and #2

This work is really impressive and has great potential but does not yet quite deliver on that potential.

The authors have used an impressive dataset of vegetation time-series in concert with plant species' functional trait and genetic information to explore trends in multiple facets of diversity over the last 100 years. They show that while species richness and phylogenetic increases at the site-level there is a concomitant decline in functional evenness. However, trends are highly influenced by other extraneous factors such as habitat type and trajectory over the time-series, indicating the nuances required to assess local biodiversity change.

The debate and controversy about the average trend in species-richness in local assemblages have been ongoing for over a decade. The lack of resolution reflects a combination of biased data and often slightly simplistic analyses. Resolving the question for such an ecologically vital and taxonomically broad group as plants, across such a broad domain as Europe, would be a major contribution on an important topic with broader societal relevance.

Irrespective of what the answer of such a reanalysis is, it would be wholly appropriate for publication in Nature Communications: it's such an ambitious and thoughtful analysis on such an important topic that I can certainly see myself - and many others - citing it for years to come. It is a very large and geographically extensive data set of plots, combined with trait and phylogenetic information; time spans are usually long enough to at least dilute the obscuring effects of short-term volatility, which plague interpretation of trends in short time series; the analyses try to not only elucidate the trends but to decompose variation among sites; and the analyses are explained clearly, making it easy to follow what was done. There is a lot to like! But, there are a few major concerns that mean I think publication of the manuscript as it is would be a mistake.

1. There is no sensitivity analysis to explore how far the patterns might have methodological rather than ecological explanations.

a: Plot area can vary by up to a factor of 2. While understanding that insisting on equal plot area might reduce the size of the data set, it would be naïve to expect a nearly twofold increase in plot area to have no effect on species richness. Conservatively taking the exponent of the species-area relationship as 0.15, plot area alone could be causing a $\pm 10\%$ "change" in species richness between consecutive surveys. Do the results change substantially when only equal-area plots are analysed? Perhaps better, what is the empirical relationship between $\log(\text{Richness}_1/\text{Richness}_2)$ and year when controlling for the effect of $\log(\text{Area}_1/\text{Area}_2)$ in the model?

b: Some plots are permanent and others are semi-permanent. Do these show the same trend? Permanent plots are conceptually much purer. Effects having an ecological basis are expected to be stronger (larger coefficients) in the permanent plots, even if they are less significant because of reduced sample sizes.

Response: We are thankful to reviewer #1 for their very positive and encouraging comments on the novelty and relevance of our work. We sincerely appreciate such recognition and we do hope that our work will be cited for years to come. We agree with reviewer #1 that our analyses, just as any synthesis across heterogeneous data sources, could be influenced by methodological differences among the synthesized resurvey projects. We now openly address this issue and reported the statistical effects of plot size, resurvey design, and size changes (Extended Data Table 5 and Fig. 6, L. 266–272, 388–390, 673–678).

2. The authors acknowledge that the data are not geographically representative but disappointingly make no effort to correct for the bias. The data are also probably not representative in terms of land-use trajectories. The reason these biases are important is that, without correcting for them, the paper simply can't estimate the true average trend in species richness in local assemblages in Europe. Without that, the set-up for the paper doesn't really work, and it's harder to see that it represents a major step forward in understanding - we're left with wondering whether the pattern we see is driven by haphazard sampling. The transition matrix between EUNIS habitats provides a way of weighting each time series, such that those for under-represented 'transitions' (which might be sites that stayed in a particular, poorly sampled, habitat type) get upweighted relative to those for over-represented 'transitions'. This would provide the best available estimate of the actual average trend in each of the diversity metrics. Please add this change!

Response: We thank the reviewer for raising this very important point. We acknowledge that our analyses, up to now, only covered average trends across the ResurveyEurope dataset and not the balanced trends across habitat types. Yet, we very much liked the suggested idea of weighting all time series for balanced average biodiversity trends and, thus, now incorporated this approach, making a distinction between “average” and “balanced” biodiversity trends (Extended Data Table 4–5, L. 254–265, 663–672).

3. Species richness and Faith's PD are both modelled, which is great, but there's no parallel analysis of any functional richness measure, which is a shame as it would be lovely to be able to compare across all three dimensions. It might also give support to currently only marginally supported statements about change in functional diversity over time.

Response: To make the trends of functional diversity indices better comparable to the trends in species richness and PD, we now included analyses of functional richness — as derived from volume of convex hulls in trait space, replacing functional dispersion which, similarly to functional divergence, quantified the spread away from the centre of trait distributions (Box 1, Figs. 3–4, Extended Data Tables 3–5, Figs. 2,3, 4, 6, L. 311–312, 321–322).

4. Not a request for more analysis, but the term “species pool” is used incorrectly throughout. Rather than being the set of species from which any of the communities in the data set might be assembled, the concept given this name in the manuscript is in fact gamma diversity for a particular habitat. Please fix this.

Response: We agree with the reviewer and accordingly replaced the term “species pool” with “habitat-specific European gamma diversity”.

Minor issues

- Captions and figures/boxes are generally not very easy to follow (though Fig 1 is nice); Box 1 places the symbols in the equations out of sequence relative to where they appear in the box itself, making it hard for the reader to join them to their equations. Figure 2's Sankey diagram is confusing (because it leaves out the sites that stayed the same - at least add the % of each habitat that do stay the same to the column at the left), and the transition lines don't align with the boxes they come from. Figure 5 would be better presenting the changes in percentage terms rather than absolute units that don't mean much to readers. Relating to this last point, the text would be improved by reporting the percentage changes too.

Response: We tried to improve the figure captions as follows. In Box 1, we re-ordered the equation symbols in order of appearance. In Fig. 2, we updated the Sankey diagram, so that the transition lines are aligned with the boxes. We would like to not add the number of stable time series, as there is already a lot of information in the figure and this data is provided in text and supplements. In Box 1, and Fig. 3–4, we now present annual percentage changes instead of absolute changes. Throughout the whole manuscript, we now focus on annual percentage changes (%) which we supplemented with the linear changes on the original units of the diversity variables (Extended Data Tables 3 and 4, L. 248–253, 261–265, 642–651).

- It is claimed (line 275) that a majority of time series show an increase in species richness, but this is not demonstrated (not the same thing as the mean change being positive).

Response: The reviewer is right, that this claim is not backed by our analyses. Thus, we changed it to “...an average increases in local plant cover, species richness, as well as number and cover of non-native species and habitat generalists” (L. 352–356).

- Line 307-310: Ms suggests that increase in TD & FD in mires and wetlands could be caused by an increase in grass dominance - but wouldn't that reduce FD (and perhaps also TD)?

Response: In the long run, we agree with the reviewer that an increase in generalist grass species should result in lower functional and phylogenetic diversity. Yet, when wetlands are initially invaded by shrubs, trees, and generalist grasses, this might first lead to the observed increase in plant diversity while deteriorating trajectories will rather play out in the future. We re-wrote this section to make this clearer: “Increases in cover and diversity of mire and wetland communities might be unrelated to conservation efforts, but rather reflect general reductions in habitat quality from desiccation and disturbances that allow the establishment of shrubs, trees, and more generalist grass species (Extended Data Figs. 3–4). Here, future resurveys are needed to evaluate whether reported increases in plant diversity could finally lead to a general decline of plant diversity in these peculiar habitats” (L. 371–379).

- The way time series are weighted (lines 718-721) is not correct.

Response: Our goal was not to conduct a classic meta-analysis for which the impact of each time series would indeed be weighted by the inverse of the standard error of its linear trend. This would be appropriate for the testing of general ecological phenomena that should (with deviations) be present in each time series. Instead, we wanted to summarize the observed trends, while giving time series with more observations just a bit higher weight. Specifically, we wanted to give time series with more observations a higher weight than time series with few observations whose linear trend has a minimal standard error.

- No real analysis of trends of generalist and specialist species, could perhaps be something easy to do to support conclusions made.

Response: We agree with the reviewer that our claims of generalist and specialist trends were rather speculative and should be backed by data. Thus, we now added analyses on the trends of habitat specialists, generalists, as well as community-weighted mean niche width, which show a significant average increase in the number and cover of generalist, but not (generally) for habitat specialists (Box 1, Fig. 4, Extended Data Tables 3–4, L. 128–134, 237–241, 248–253, 600–614, ...). We would like to

thank the reviewer for suggesting these additional analyses as they now nicely support our claims of positive trends for generalist species.

- Taxonomic harmonization to a list of species compiled recently will lead to an upward bias in species numbers over time. It would be much better to harmonize to an old list, so that entities newly recognised as species between surveys are treated consistently.

Response: Here, we do not completely agree with the reviewer as taxonomic harmonization can lead both to higher and lower species richness, as new species could be recognized but also, in quite some cases, species that were considered valid in the past are now considered synonyms of other species. We put every effort into cleaning the dataset of taxonomy-driven changes in species numbers (L. 540–550), even conferring with the original data owners to reduce pseudo-turnover or spit of species to different resurveyors. Apart from that, we have no way to harmonize novel data to old taxonomies and would argue that the remaining artifacts, as far as they exist, cannot have a significant impact on our results.

- The writing around threatened species needs to be a bit clearer. The list of which species are threatened is static, so having more threatened species in an assemblage is likely to be indicative of a recovery rather than a general decline. The authors clearly understand this but not all readers will.

Response: We fully agree with the reviewers interpretation of increases in Red List species. To better translate this into our manuscript, we noted that we used a static Red List dataset (L. 214–216, 604) and specifically explained how the observed increases in Red List species should be interpreted: “As the latter was determined from static Red Lists, the observed increases in Red List species refers to an increase in threatened species and not to altered threat classifications” (L. 202–203, 244–248).

Minor points indicated in the manuscript file

Response: We incorporated all the indicated improvements to grammar and corrected the highlighted spelling mistakes. Below, we responded to the specific comments reviewer #1 made in the manuscript file.

Line	Comment (summarized)	Our Response
112	More meaningful as percentage	We agree and shifted the focus of the results to percentage changes.
114	increases?	We replaced “strongest” with “most pronounced”.
117	That's NOT a species pool	We agree and changed the wording to “habitat-specific European gamma diversity”.
119	If it's the same thing as species richness, then line 112 needs immediate clarification	We added that we were discussing “habitat-specific gamma diversity” to avoid confusion with species richness.
123	Awkwardly worded.	We re-worded this introductory sentence to “Losses of plant diversity can alter the health and functioning of ecosystems, with negative effects also for human well-being” (L. 144–145).
124	worldwide	We replaced “at the global extent” with “worldwide”.
135	Better to say "not just for taxonomic richness (i.e....) but also in terms of functional diversity, phylogenetic diversity and extinction risk."	We altered this sentence to “This is the case not only for taxonomic diversity, but also the diversity of” (L. 156–161).
143	Can delete these words	Deleted accordingly.
151	we have mobilised existing time series of vegetation plots and compiled them across large...	Changed to “we have mobilized existing vegetation-plot time series data...” (L. 173–174).
155	That's not a meaningful species pool.	We agree and replaced “species pool” with “habitat-specific European gamma diversity”.
158	allow	Replaced with “to quantify” (L. 180).
171	change	Altered to “. Even within relatively stable time series (e.g., plots that remain forested), a change of climate, pollution, and management practices can lead to likewise

		predictable shifts in plant composition" (L.95–197).
173	Give some of the info in lines 201-204 here to forestall reader questions	As suggested, we moved the information on the number of observations and time spans covered to this section.
177	Did you classify the sites for the years in between the surveys as well as at the time of the surveys?	We only focussed on habitat changes between the first versus the last observation date and made this clearer "Based on the assigned level 3-habitat types at the first versus last observation date..." (L. 207–210).
184	Not a sensible hypothesis	We agree and thus toned down the explanation of our hypothesis to "...negative trends in functional and phylogenetic diversity due to the replacement of distinct habitat-specialist with more similar generalist species" (L. 218–220).
192	largest RELATIVE increases?	Replaced with "comparatively largest increases" (L. 256).
200	Histograms would be good	We agree and added histograms in Extended Data Fig. 1.
208	much more interpretable as percentages	We agree and now focussed on percentage changes throughout the manuscript.
219	How? This is opaque.	We now clarified that we developed this classification scheme of habitat trajectories "Based on the EUNIS level 3 habitat type at the first and the last observation date, we assigned all time series into our own scheme of habitat trajectories"
221	23.4% is not a majority!	This is right and we thus removed the hint to the majority.
258	Do I understand correctly that the list of which species are threatened was static across all years?	Yes, and we now make this clearer throughout the manuscript (L. 202–203, 244–248).
274	You have shown an increase ON AVERAGE, not that the MAJORITY have increased. Have they?	This is right and we accordingly re-worded to "... no consistent continental decline of habitat-specific gamma diversity but, on average, an increase in local plant cover, species richness,..." (L. 207 – 217).
285	These are pretty big caveats to introduce at this point! Are any of the data immune to these issues.	Indeed, vegetation-plot time series come with a lot of potential caveats, but we wanted to openly address them. Additionally, we now also showed the statistical effects of permanent versus semi-permanent plot design, plot size, and plot-size changes in Extended Data Table 5 and Extended Data Fig. 6.
287	Could re-surveys have been using different floras containing more species?	No, as taxa were harmonized when time series were added to the European Vegetation Archive.
291	Why not model species losses and gains separately here? (and indeed what is the general compositional turnover between surveys, and the level of sampling completeness that that implies?)	Although this is an interesting and important question, the ration of losses and gains has been addressed elsewhere (e.g., Jandt et al. 2022, Nature, DOI: 10.1038/s41586-022-05320-w). Adding a completely novel field of compositional turnover would likely blow up the manuscript but is planned to be addressed in future analyses of the MOTIVATE project.
292	This is all fine but means that the data aren't spatially or ecologically representative.	Yes, as with all resurvey data, the are just a collection of scientific data that has been collected for varying purposes. However, we now supplemented our results with balanced diversity analyses in which each EUNIS level 3 habitat type contributed equally to partly account for unbalanced data.
317	They are not "comprehensive"	Indeed. Our dataset is not comprehensive, but the most comprehensive (up-to-date). We replaced comprehensive with "massive dataset" (L. 403).
321	at least partly!	Yes, which is why we highlighted that the obtained trends could, at least in part, be biased. Additionally, we now also included sensitivity analyses on resurvey design, plot size changes, and balanced diversity trends to make sure that our main findings of increases in biodiversity are generalizable.
332	more generally, ANY methodological inconsistency	We agree and changed the wording to "generally be checked for methodological inconsistencies, including ..." (L. 412 – 413).
Fig. 1	Several issues: Plot area can change by a factor of up to 2! Taxonomic harmonization means not respecting surveyors' opinions of what's distinct; it's gamma richness not species pool richness; how were linear regressions run when N=2?	We addressed all of these issues as follows: (1) We added analyses on the impacts of plot size changes on observed diversity trends. (2) As outlined, we took as much care as we could to not distort our analyses by taxonomic harmonizations. (3) We changed all terms to "gamma diversity". (4) We explained how linear regressions are to be interpreted when based on only 2 observations (L. 652–655).
Fig. 2	How was EUNIS used for 1940?	As outlined in the text, each vegetation-plot observation was assigned to EUNIS habitats based on location and species composition.
Fig. 2	How do changes in habitat type correspond to the Europe-wide estimate of how much land has undergone each transition? (i.e., how representative?)	This is rather out of the focus of this study. Also, we would be careful to not scale our habitat-specific diversity trends to the European scale (only for habitat-specific gamma diversity). Monitoring of general trends across European regions is not (yet) feasible with scientific time series only.
Fig. 2	NOTHING IS STILL A GRASSLAND!!!	This is correct and reflected in the panel label "Changes in habitat types". This figure only shows the shifts that occurred in EUNIS level 1 habitat types. As stated in the figure caption: "Time series with stable habitat types and time series that could not be assigned to a habitat type were omitted from panel c (but are shown in Supplementary Figs. 3–4 ...".
Fig. 4	Would be much more meaningful to show the responses as changes relative to the initial value	We agree and changed the focus of the manuscript on annual percentage changes.
Fig. 5	Good to have taken care of the	We appreciate the appreciation.

	'expanding survey' bias by considering decade-to-decade changes at sites.	
592	Does this make a difference, empirically?	We now quantified the statistical difference between semi-permanent versus permanent resurvey designs in Extended Data Table 5.
603	Too permissive! At least needs a sensitivity analysis here.	We now added analyses on the statistical impact of changes in plot sizes in Extended Data Table 5.
613	???	We tried to make the problem with N-to-N relationships more clear by re-phrasing as “when older plots could not be exactly re-located so that the re-sampling was conducted in proximate representative habitats and with sometimes differing plot numbers so that diversity trends had to be estimated in these so-called N-to-N relationships of semi-permanent plots” (L. 523–526)
626	If taxonomies increased in length over time, the trend in plot-level richness will be biased upwards.	We see the reviewers caution of “pseudo-splits” in species...which is exactly why we did these tests for potential name mismatches. We hand-checked hundreds of time series for any obvious pattern of pseudo-turnover, pseudo-splits, or pseudo-harmonization...even conferring with original data providers to make sure that our analyses are as unbiased as possible (with regard to taxonomic harmonizations).
632	lots of arbitrary choices here	The choices of minimum plot numbers and thresholds were set very low so that a large number of time series were checked by hand to make sure that we found every suspicious pattern of pseudo-turnover or pseudo-losses/gains.
650	why not 4895? (line 542)	This number of species refers to those species for which we had data in the gap-filled trait dataset. We tried to highlight this better as “... and extracted the subset of 3,834 taxa that occurred in our time series of vegetation plots” (L. 566).
703	Why no sign test here? (cf line 756)	As we were interested in the general trends across and not within individual time series, there was no need for significance tests + as most time series had only 2–5 observations, these tests would usually be non-significant. We tried to highlight better that these models were just used to summarize diversity trends to one value per time series as “Time series-specific trends in diversity indices” (L. 641).
703	For both S and PD, more natural to analyse on a log scale (or in a GLM with Poisson errors) and express results relative to the intercept	We agree that the natural log would be a better choice if we would attempt to model species richness trends as correctly as possible (to account for non-normal error distributions). We did not include this in our study for two reasons. (1) We now focus on percentage changes for which such a correction is not necessary and (2) for the supplementary analyses on changes in original diversity units, such a correction does not improve our trend estimates when time series have only 2–5 observations.
717	But we know how s.e. scales with N, and it's not as the log.	This is correct, but, as outlined in the general responses above, our goal was not to conduct a classical meta-analysis on the relationship between year and diversity but to calculate average trends across all time series for which such a weighting scheme cannot be applied (especially not for time series with only two observations).
757	I think these are more meaningful than the absolute changes and should be presented in preference	We agree and thus shifted the focus of the manuscript on annual percentage changes.

Reviewer #3

The manuscript entitled “Habitat-specific trends in taxonomic, functional, and phylogenetic diversity in European plant communities during the last 100 years” by Kambach et al. provides a large-scale analysis of vegetation time series across Europe, quantifying general and habitat-specific trends in taxonomic, functional, and phylogenetic plant diversity, alongside changes in Red List species. The authors report a general increase in species diversity but contrasting response patterns across habitats. The manuscript stands out for the articulation and harmonization of vegetation time series derived from phytosociological surveys, which can synthesize the trajectory of local co-occurrence of plant species across Europe. This type of information cannot be obtained from satellite products or from the distribution of individual species alone, which makes the results of this manuscript particularly relevant. Overall, this is a high-quality contribution, but I have some concerns regarding a key point, which I summarize below. Additional minor comments are provided in the annotated text.

A. Main concerns regarding the hypotheses and their implications

(1) Replacement of specialists by generalists

The authors state that local plant communities show stable species richness but negative trends in functional and phylogenetic diversity due to the replacement of habitat-specialist “loser” species with more generalist “winner” species sharing similar ecological strategies and evolutionary histories (8, 38, 39).

It is not clear to me how the authors tested and demonstrated that the winners are indeed generalists with similar ecological strategies and evolutionary histories. Additionally, I assume that the specialist/generalist classification was applied only to native species; otherwise, this could create confusion, particularly in relation to hypothesis 3. If the winners are native generalists, I would indeed expect similar ecological strategies and evolutionary histories, but if they are non-native species, I would expect the opposite pattern at least for the evolutionary history.

Response: We agree with reviewer #3 (whose point here was also raised by reviewers #1 and #2) and, thus, added analyses on the trends of habitat specialists and generalists as well as non-native species to the manuscript. As species-specific niche-width estimates were derived from species co-occurrence in the ReSurvey Europe dataset, we were able to also include non-native species. Our results clearly show a positive trend for the number and cover of generalist and non-native species (Box 1, Figs. 3–4, Extended Data Tables 3–5, L. 130–134, (H3) 237–248, 596–614, ...).

(2) Habitat types and habitat-change trajectories

The authors hypothesize that differences in local diversity trends are related to both habitat types and habitat-change trajectories, expecting the highest gains in sparsely-vegetated habitats undergoing succession and the highest losses in specialized wetland and forest habitats undergoing disturbance. References would be useful here to support this statement.

Response: We agree with the reviewer that our claims of highest gains in sparsely-vegetation and specialized wetland habitat are not backed by studies that compare the diversity trends among the number of habitat types that we investigate. Thus, we toned down this hypothesis to “(H2) Differences in diversity trends are to be expected due to the different habitat types, habitat-change trajectories, and observation dates” (L. 220–222).

However, with regard to forest habitats, I do not fully agree with the authors’ interpretation. Late-successional forests generally host fewer species than early-successional states due to the strong competitive “umbrella effect,” which typically allows only a few shade-tolerant species to persist. High species richness in late-successional forests is usually restricted to tropical forests in large plots. Similarly, in grasslands there is evidence that increased biomass and light limitation are associated with richness declines (e.g. Borer et al., 2014, Nature). Thus, as a general hypothesis, one could state that successional trajectories may be associated with richness declines, while disturbance trajectories may be associated with richness increases.

Response: We agree with the reviewer that plant diversity should be higher in early successional as compared to late-successional/climax forest or grassland habitats. Yet, apart from evidence suggesting the highest plant diversity in European climax forests (e.g., Hilmers et al. 2014, J Appl Ecol), one could also follow the intermediate disturbance hypothesis which suggests that the highest plant diversity should generally be found in moderately disturbed communities with the highest structural complexity. Thus, we know that our classification of successional versus disturbance trajectories might not always be ecologically the most plausible, but to be able to generalize, we

specifically stated that we regarded succession-associated shifts as those towards higher biomass and/or habitat complexity.

More broadly, in this hypothesis the authors attempt to incorporate two factors, habitat type and habitat-change trajectory, which is also one of their main results. However, it is difficult to interpret these factors together and to integrate this hypothesis with Fig. 4 and the main text.

Response: To make H2 more specific, we altered it to “Differences in diversity trends are to be expected due to the different habitat types, habitat-change trajectories, and observation dates”. We further expanded the variance partitioning approach shown in Fig. 3 to incorporate the interaction between the habitat-type and habitat change trajectory; which emerged as the best predictor, thus highlighting the context-dependency of local biodiversity trends (L. 300–305).

Finally, I am not sure if the authors are referring to particular forest and wetland types when writing “specialised wetland and forest habitats.” If so, this should be specified, since no internal classification of habitats is provided in the manuscript.

Response: We acknowledge that we did not provide supplementary analyses on “specialised habitats”. However, to account for trends in habitat specialists versus generalists, we now added these analyses, showing a general increase of habitat generalists but not specialist species (Box 1, Fig. 4, Extended Data Tables 3–4, L. 128–134, 237–241, 248–253, 600–614, ...).

(3) Expansion of the European species pool

The authors argue that species pools have increased over the last century due to the spread of non-native and generalist species, and hypothesize that man-made and disturbed habitats will show the largest increases, since these habitats are more susceptible to invasions (40, 41). I agree with this hypothesis. However, how can the reader distinguish whether the winners are native generalists or non-native species? This seems problematic, especially since the conclusions attribute the main changes to non-native species.

Response: We agree with the reviewer and thus added analyses, showing that local plant diversity showed significant increases in non-native and habitat-generalist species (Box 1, Fig. 3, Figs. 2–4, Extended Data Tables 3–4, L. 128–134, 237–241, 248–253, 600–639, ...).

In addition, the hypotheses do not explicitly mention threatened species. Should we expect any particular patterns for them? Or is the assumption that threatened species are primarily specialists?

Response: We did not add a particular hypothesis on threatened species, as we think that their observed local trends depend on the nature of the species, and what makes them rare. Yet, we now supplemented a small overview on the relationship between species’ niche width and Red List status, showing that threatened species have slightly, albeit significantly, lower niche width estimates with $R^2 = 0.006$ (Supplementary Box 1, L. 244–248).

B. General observation

I found a partial but important disconnection between the hypotheses and the results, as well as a strong focus on richness patterns compared to the other response variables. The hypotheses raise

strong expectations that are not fully met in the results and discussion. I encourage the authors to present their findings in a more comprehensive way that more directly addresses their hypotheses.

Response: We acknowledge the reviewer’s critique that the tests of our main hypotheses had been somewhat buried among the many results and diversity trends in our manuscript. To make it easier for the reader to grasp our main findings on these hypotheses, we now explicitly refer to them throughout our result section in the following lines sentences.

“Considering these findings, we have to reject our hypotheses of a generally stable or decreasing local plant diversity in Europe (H1)” (L. 243–244).

“In accordance with our hypothesis (H2), we found that the proportion of explained variation in diversity trends was, on average, most tightly related to interactions among level 3 habitat type and habitat-change trajectory ...” (L. 300–303).

“In contrast to our hypothesis (H3), European gamma diversity, quantified as the summed number of unique taxa across all time series, did not follow a general or any temporal trends (Extended Data Fig. 5)” (L. 337–339).

Minor points indicated in the manuscript file

Response: We incorporated all the indicated improvements to grammar and corrected the highlighted spelling mistakes. Below, we responded to the specific comments reviewer #3 made in the manuscript file.

Line	Comment (summarized)	Our Response
197	Check format of the subtitles?	We checked and adapted the subtitle format to meet the Nat Com style.
270	Yes, but this may be due to an increase in non-native species. This is a really important detail that you must analyse in order to discuss your results properly.	We agree and thus added analyses on the trends for non-native species throughout the whole manuscript, showing that non-native species significantly increased across time series (Box 1, Fig. 3, Figs. 2–4, Extended Data Tables 3–4, L. 128–134, 237–241, 248–253, 600–639, ...).
279	But you don't have the information to support this statement! You must analyse this. It is directly linked to hypothesis 2.	Reviewer #3 is right here, that our interpretations were not based on data, which is why we added analyses on non-native, specialist, and generalist species, which comply with our interpretations (Box 1, Fig. 3, Figs. 2–4, Extended Data Tables 3–4, L. 128–134, 237–241, 248–253, 600–639, ...).
336	But you found a general positive trend in the local pool, didn't you? Or are you referring to the trajectories across different habitats?	This is right and we changed this first sentence of the conclusions to “Synthesized over a century of European vegetation-plot time series, we detected prevailing positive trends the diversity of local plant communities” (L. 420–421).
339	Did you analyse the native and non-native groups separately? In what way do your results support this statement?	Adding analyses on trends in non-native species, we can show that non-native species significantly increased in our time series. However, since we did not separately analyse the trends for native species, we specified that we observed decreasing trends in habitat specialists (and not in native species, L. 423–424).
340	Why? Based on Fig. 2, these habitats have already been transformed into forests, shruland or grassland. I think these results are in line with previous studies. Am I right?	As noted in the caption, Figure 2 shows only those time series with changes in EUNIS level 1 habitat types, whereas all trajectories (including stable ones) are shown in Supplementary Fig. 3–4. Yet, in Extended Data Fig. 3, we also demonstrated that time series in inland sparsely and vegetated man-made habitats generally showed strong trends in most diversity indices.
345	Could you include some future trends or expectations based on your results?	Although we understand that some kind of prediction on anticipated trends would be great to end with, we would like to refrain from speculations, especially in consideration of the reported interdependence of diversity trends on the time period, habitat type, and habitat-change trajectory. This is why we now specifically state “Still, due to the complexity of ecological systems, also with regard to the intricate effects of climate and land-use change, the site- and habitat-specific future trends in local plant diversity cannot yet be reliably extrapolated from the compiled vegetation-plot time series” (L. 427–430).
Box 1	Change “red list species to threatened species” here	In favour of a consistent terminology, we decided to call this group of species “threatened Red List species” throughout the manuscript and “Red List species” in the figures and tables.
Fig. 1	Included the Open Tree of Life into the box of “data sources”	As correctly noted, we added the missing reference to the Open Tree of Life (Fig. 1).

Fig. 2	This is probable a extreme minor details, But my mind associates green with forest and yellow with grassland. Could you change these colors?	For clearer interpretation, we adapted the suggested colour scheme throughout the whole manuscript, extended, and supplementary figures.
Fig. 3	Change “red list species to threatened species” here and in the other figures.	Similar to our response to the comment to Box 1, we would like to keep the terminology of “Red List” or “threatened Red List” species.
Fig. 4	This is probably another minor detail. Could you change 'circulate' to 'triangle' in the legend?	As suggested, we altered the symbols in the legend, her to a diamond shape which we think better reflects that trends could be positive (upward triangle) or negative (downward triangle).
600	A small details, but important to me is that EUNIS level 1,2 o 3 it's too abstract for someone not familiar with this system. Could the authors bright an example of the differences level in methods?	To give the readers that are not be familiar with the EUNIS hierarchy of habitat types a better introduction, we now give examples of the different EUNIS levels as “... ranges from broadly defined EUNIS level 1 habitats (e.g., T - forests) to more narrowly-defined level 2 habitats (e.g., T1 - Deciduous broadleaved forest) and level 3 (e.g., T18 - Fagus forest on acid soils)” (L. 185–187).
602	This eunis level was using in fig. 3 but it has was effectively assigned to the 34% of the data!!	It is correct that only 34% of time series could be assigned to EUNIS level 3 habitat types which we now state more clearly as “For a subset of 8,764 time series with sufficient replication of EUNIS level 3 habitat types and trajectories (see Methods)” (L. 292–293).
609	I am worried about the difference in eunis level, because I think most of the data were finally classified in eunis 1 o 2. What are de diferencias between level?? Could you mention an example? I think it could be important for a wider audience not familiar with eunis system.	Similar to our response to the comment in L. 600, we now provide the reader with an example of the EUNIS hierarchy of habitat types as “..., ranging from broadly defined EUNIS level 1 habitats (e.g., T - forests) to more narrowly-defined level 2 habitats (e.g., T1 - Deciduous broadleaved forest) and level 3 habitats (e.g., T18 - Fagus forest on acid soils)” (L. 186–188).
609	However, as you mentioned above most of the data could not be classified at EUNIS level 3. So did you use only EUNIS level 3 or alternative you also use the other levels?	We now specify that “unclassifiable time series could not be assigned to EUNIS level 3 habitat types” (L. 216–217).
667	All the species were clasified? At least in other part of the world there are many species that in generar are not mentioned in red list.	We now specify better that we classified all species as Red List species “species if they were classified as “threatened” on either the European Red List or on the national Red Lists of the country in which the vegetation plot was situated” (L. 593–595). Otherwise “we classified species as “not threatened” when they were either not listed in the Red Lists or listed as Near Threatened (NT) or Least Concern (LC)” (L. 591–593).
675	This happen in most of the survey? It is rare to me...	This only happened in a minor fraction of vegetation-plot observations but, in our experience, usually has to be adapted when bringing together different data from different resurvey projects.
735	Claro, re grosso... osea que además de diferencias en el tamaño de plot de different time series esto es también complicado...	Si 😊. Semi-permanent plot with different number of resurveys located proximate to the historic vegetation plot is common in large vegetation-plot time series. We rephrased this section to make the problem with N-to-N relationships clearer as “. Time series-specific diversity trends could cover multiple vegetation plots, e.g., in case of N-to-N relationships from semi-permanent resurvey plots for which one or more original vegetation plots were resurveyed with several new plots in proximate locations in the same habitat instead of specific plot locations” (L. 657–660).

Reviewer #1 (Remarks to the Author):

I thank the authors for having engaged so thoroughly and thoughtfully with my comments and suggestions. This revised version of what was already a very impressive manuscript has allayed all my major concerns and also dealt entirely satisfactorily with all the minor points I raised. As I said in my initial review, I think this improved version absolutely meets the required novelty, ambition and rigour for publication in Nature Communications, and I very much look forward to citing it!

Reviewer #2 (Remarks to the Author):

Response: We are very happy with the support from Reviewer #1/2 and would like to thank them for their constructive critique.

Reviewer #3 (Remarks to the Author):

I really enjoyed reading the revised version of this manuscript and I appreciate the authors' detailed responses and the substantial improvements made. I have only one remaining comment.

In lines 232–235, the authors state that, because disturbed and nutrient-rich habitats tend to be more susceptible to invasive species, they expected time series from disturbed and man-made habitats to show comparatively larger increases in habitat-specific gamma diversity. However, this expectation is not explicitly revisited in the Results/Discussion (lines 345–356). As the analyses show largely non-significant responses across habitats and trajectories, and even negative trends in stable trajectories and successional forests, a deeper interpretation may indeed be too speculative. Nevertheless, it would be helpful to explicitly acknowledge that this expectation was not supported by the results.

One possible explanation could be that recent European landscape trajectories have often been characterized by forest expansion and a reduction in disturbance intensity (e.g. grazing abandonment), which may limit the spread of non-native species. This would contrast with patterns reported from regions experiencing forest degradation and rapid urban expansion, where increases in non-native species are more commonly observed.

Response: As correctly pointed out by reviewer #3, we did not discuss our expectation of increasing gamma diversity in disturbed and man-made habitats. We added that this expectation was not met in L. 350–352. We agree that deeper interpretations and discussions of these patterns would have been too speculative. Also, we cannot agree with the interpretation that lower gamma diversity in forest and higher gamma diversity in disturbed habitats might be related to forest expansion or reduced intensification because our results can in no way account for patterns of forest expansion or land-use abandonment, simply because the scientific time series are not spatially representative.